# RNA Polymerase II transcription independent of TBP in murine embryonic stem cells

**James ZJ Kwan**[1†]**, Thomas F Nguyen**[1†]**, Anuli C Uzozie**[2,3]**, Marek A Budzynski**[1]**, Jieying Cui**[1]**, Joseph MC Lee**[1]**, Filip Van Petegem**[1]**, Philipp F Lange**[2,3]**, Sheila S Teves**[1]***

[1]Department of Biochemistry and Molecular Biology, Life Sciences Institute, University of British Columbia, Vancouver, Canada; [2]Department of Pathology and Laboratory Medicine, University of British Columbia, Vancouver, Canada; [3]Michael Cuccione Childhood Cancer Research Program, BC Children's Hospital Research Institute, Vancouver, Canada

**Abstract** Transcription by RNA Polymerase II (Pol II) is initiated by the hierarchical assembly of the pre-initiation complex onto promoter DNA. Decades of research have shown that the TATA-box binding protein (TBP) is essential for Pol II loading and initiation. Here, we report instead that acute depletion of TBP in mouse embryonic stem cells has no global effect on ongoing Pol II transcription. In contrast, acute TBP depletion severely impairs RNA Polymerase III initiation. Furthermore, Pol II transcriptional induction occurs normally upon TBP depletion. This TBP-independent transcription mechanism is not due to a functional redundancy with the TBP paralog TRF2, though TRF2 also binds to promoters of transcribed genes. Rather, we show that the TFIID complex can form and, despite having reduced TAF4 and TFIIA binding when TBP is depleted, the Pol II machinery is sufficiently robust in sustaining TBP-independent transcription.

**\*For correspondence:**
sheila.teves@ubc.ca

[†]These authors contributed equally to this work

**Competing interest:** The authors declare that no competing interests exist.

## Editor's evaluation

This important study employs auxin-induced degradation to provide compelling evidence that the TATA-binding protein (TBP) is not required for ongoing RNA polymerase II transcription nor heat-shock or retinoic acid-induced transcription in murine embryonic stem cells, but is essential for RNA polymerase III transcription. Furthermore, the study shows that TBP-independent TFIID complexes are assembled and present at the transcription start sites of polymerase II-transcribed promoters. This work will be of broad general interest to the regulation of gene expression field.

## Introduction

Transcription is the first step in gene expression, and it begins with the stepwise assembly of general transcription factors (GTFs) and RNA Polymerase II (Pol II) to form the pre-initiation complex (PIC) at the promoters of genes (*Matsui et al., 1980*; *Reinberg et al., 1987*; *Buratowski et al., 1989*). First, the GTF TFIID, composed of the TATA-box binding protein (TBP) and 13–14 evolutionarily conserved TBP-associated factors (TAFs), binds onto gene promoters (*Reinberg et al., 1987*; *Buratowski et al., 1989*; *Dynlacht et al., 1991*; *Cormack and Struhl, 1992*; *Tanese et al., 1991*), which triggers the recruitment of TFIIA and TFIIB (*Reinberg et al., 1987*; *Buratowski et al., 1989*). Pol II is then loaded

onto the promoter along with TFIIF (*Reinberg et al., 1987*; *Buratowski et al., 1989*; *Buratowski et al., 1991*). The binding of TFIIE and TFIIH completes the PIC, upon which the enzymatic activity of TFIIH unwinds DNA to form a transcription bubble, allowing for Pol II elongation and the incorporation of nucleotides into the nascent RNA (*Reinberg et al., 1987*; *Buratowski et al., 1989*; *Rimel and Taatjes, 2018*).

Decades of research have revealed a central role for TBP in the transcription initiation process. TBP was first discovered to recognize and bind to the TATA-box (TATAAA consensus sequence) in gene promoters in vitro (*Nakajima et al., 1988*; *Hoey et al., 1990*; *Buratowski et al., 1988*), though subsequent studies show that TBP binds to active gene promoters in vivo regardless of the presence of a TATA sequence (*Kim and Iyer, 2004*; *Pugh and Venters, 2016*). Early studies in *Saccharomyces cerevisiae* expressing TBP mutations show that the loss of functional TBP led to decreased GTF and Pol II recruitment and transcription in vivo (*Klages and Strubin, 1995*; *Kim et al., 1994*). More recently, acute TBP depletion in yeast cells via the anchor-away system led to a rapid, genome-wide loss of Pol II and GTF occupancy (*Petrenko et al., 2017*; *Petrenko et al., 2019*). In addition to facilitating Pol II transcription, TBP is also involved in at least two other RNA polymerases found in eukaryotes, and conditional inactivation of TBP in yeast led to a global decrease in both RNA Pol I and III transcriptional activity (*Cormack and Struhl, 1992*; *Kwon and Green, 1994*; *Schröder et al., 2003*; *Shen et al., 1998*; *White et al., 1992*). In vitro experiments using purified proteins from yeast and human cells also confirm the role of TBP in the three main RNA polymerases (*Goodrich and Tjian, 1994*; *Burley and Roeder, 1996*). However, mouse embryos harboring a homozygous TBP knock-out, which were viable up to ~40 cell stage of embryogenesis, still displayed normal levels of Pol II transcription (*Martianov et al., 2002*). Furthermore, acute depletion of TBP in mouse embryonic stem cells (mESCs) showed that global transcription levels remained unaffected (*Teves et al., 2018*), raising the question of how Pol II transcription occurs in the absence of TBP in these cases.

Several potential mechanisms may facilitate TBP-free Pol II transcription. One, TBP may be required only for the initial round of activation, but ongoing transcription may no longer needed for TBP. The impairment of the reactivation of genes following mitosis when TBP is depleted (*Teves et al., 2018*) would support this mechanism. Two, metazoan-specific TBP paralogs could functionally replace TBP. In some species, TBP paralogs have been found to bind onto genomic loci distinct from TBP and regulate cell type-specific genes (*Akhtar and Veenstra, 2009*; *Kedmi et al., 2014*). Furthermore, TBP-related factors TRF1 and TRF2, first discovered in *Drosophila,* are also able to substitute for TBP in TATA transcriptional systems (*Crowley et al., 1993*; *Hansen et al., 1997*; *Rabenstein et al., 1999*; *Holmes and Tjian, 2000*). A third vertebrate-specific paralog, TRF3, mediates transcription of oocyte-specific genes through a non-canonical complex with TFIIA/TFIIB (*Yu et al., 2020*). Three, a TAF-containing complex could facilitate Pol II transcription without TBP. Indeed, several TAF complexes have been identified, including the TBP-free TAF-containing complex (TFTC) isolated from human cells that can support in vitro Pol II transcription without TBP or TFIID (*Wieczorek et al., 1998*). TFTC was subsequently shown to correspond to the SAGA (Spt-Ada-Gcn5 acetyltransferase) complex, a conserved eukaryotic co-activator with histone acetyltransferase activity (*Cheon et al., 2020*), which has been shown to act somewhat redundantly with TFIID in yeast (*Donczew et al., 2020*).

In this study, we examined the role of TBP in transcription initiation in mESCs using an acute depletion system. We report that, inconsistent with a requirement for TBP in Pol II-dependent transcription, rapid depletion of TBP has no global effect on nascent RNA levels and Pol II chromatin occupancy. In contrast, tRNA transcription and Pol III chromatin occupancy are severely impaired upon depletion of TBP. We found that gene induction via the heat shock (HS) response and retinoic acid (RA)-mediated differentiation occurs normally upon TBP depletion. We also report that although the TBP paralog TRF2 binds to active gene promoters, it does not functionally replace TBP in Pol II transcription. Lastly, we show that the TFIID complex can form when TBP is depleted, although subunits have altered DNA occupancy. We find that TFIIA also has reduced binding when TBP is depleted. Taken together, these results point to a TBP-independent mechanism for Pol II transcription.

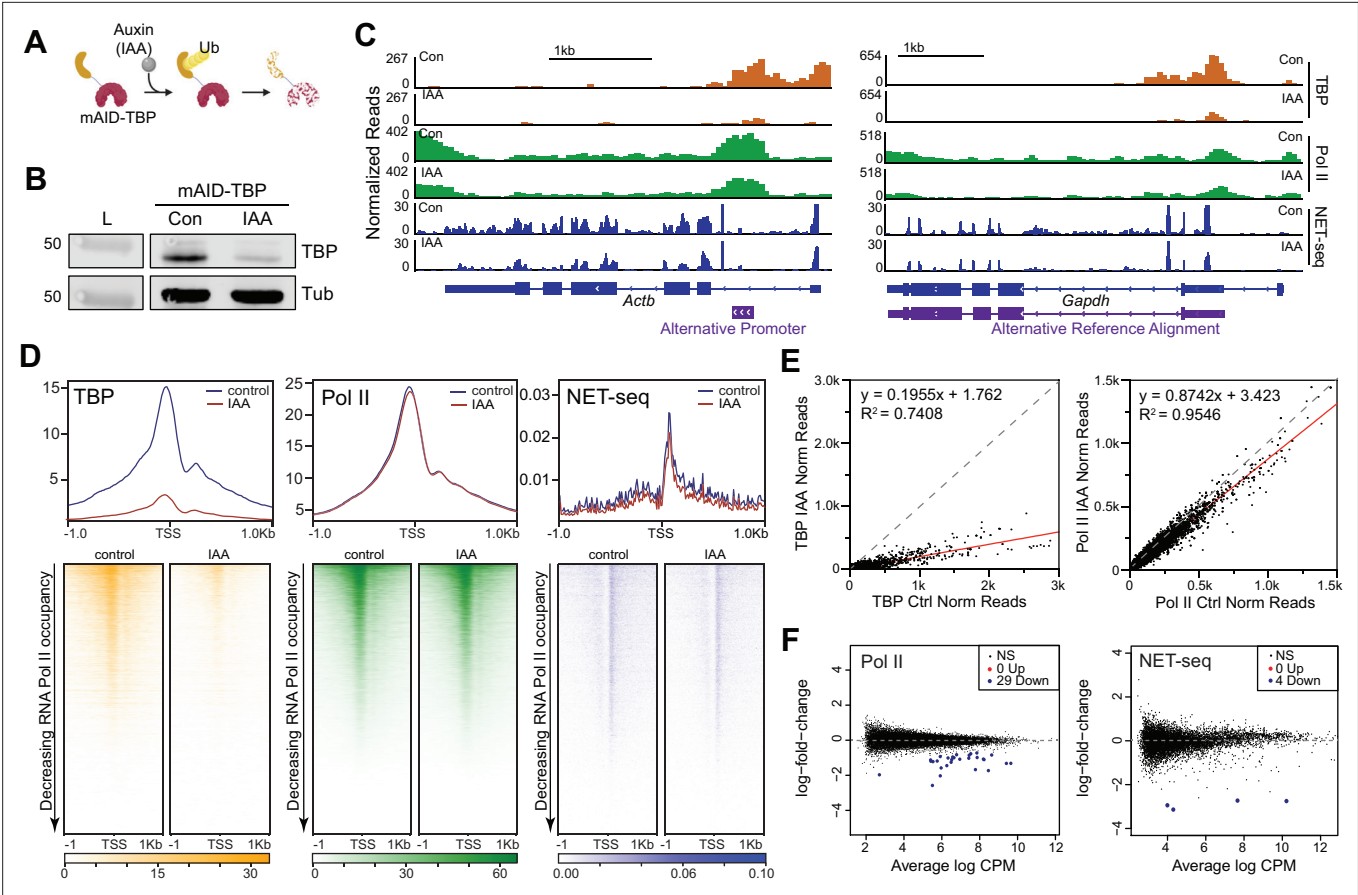

**Figure 1.** Global Polymerase II (Pol II)-mediated transcription is TATA-box binding protein (TBP)-independent in mouse embryonic stem cells (mESCs). (**A**) Schematic of indole-3-acetic acid (IAA)-mediated degradation of mAID-TBP. (**B**) Western blot analyses of whole cell lysates for control and IAA-treated C64 mESCs blotting for α-TBP, with α-Tubulin used as a loading control. Cells were incubated with DMSO (control) or 500 μM IAA for 6 hr unless otherwise stated. (**C**) Gene browser tracks of *Actb* (top) and *Gapdh* (bottom) for cleavage under targets and tagmentation (CUT&Tag) analyses of TBP (orange) and Pol II (green), and strand-specific reads from NET-seq data (blue) in control or IAA-treated C64 mESCs. Alternative promoters are indicated. (**D**) Genome-wide average plots (top) and heatmaps (bottom) arranged by decreasing Pol II occupancy of TBP CUT&Tag (left), Pol II CUT&Tag (middle), and NET-seq (right) in a 2 kb window surrounding the transcription start site (TSS) of all genes for control and IAA-treated C64 mESCs. (**E**) Normalized read counts of TBP (left) and Pol II (right) CUT&Tag signal for control vs. IAA-treated C64 mESCs in the promoter (–250 to TSS) region of each gene are displayed as a scatter plot. (**F**) Differential gene expression (DGE) analysis of Pol II CUT&Tag (left) and NET-seq analysis (right) in control vs. IAA-treated C64 mESCs.

The online version of this article includes the following source data and figure supplement(s) for figure 1:

**Source data 1.** Full uncropped membrane scans of immunoblot analyses with α-TBP control and indole-3-acetic acid (IAA)-treated C64 whole cell lysates without and with annotations.

**Source data 2.** Full uncropped membrane scans of immunoblot analyses with α-Tubulin for control and indole-3-acetic acid (IAA)-treated C64 whole cell lysates without and with annotations.

**Figure supplement 1.** Characterization of TATA-box binding protein (TBP) degradation.

**Figure supplement 1—source data 1.** Full uncropped membrane scans of immunoblot analyses with α-TBP and α-H3.3 for control and indole-3-acetic acid (IAA)-treated C64 cell lysates (input), cytoplasmic fractions (cyto), and chromatin (pellet) without and with annotations.

**Figure supplement 2.** Replicate analysis of Polymerase II (Pol II) cleavage under targets and tagmentation (CUT&Tag) and NET-seq data.

**Figure supplement 3.** TATA-box binding protein (TBP) is not required for transcription of specific subsets of genes.

## Results
### TBP is not required for global Pol II transcription
We have previously generated homozygous knock-in of the minimal auxin-inducible degron tag at the N-terminus of TBP (mAID-TBP) in mESCs (C64 cells) (**Teves et al., 2018**), which enables rapid and

acute depletion of the tagged protein upon the addition of indole-3-acetic acid (IAA) (*Figure 1A*; *Holland et al., 2012*). Treatment of C64 cells with 6 hr of IAA (+IAA) resulted in no major effects on cell viability (*Figure 1—figure supplement 1A*) with over ~90% depletion of mAID-TBP as detected by immunofluorescence (*Figure 1—figure supplement 1B*), by western blot of whole cell lysates (*Figure 1B*) and of subcellular fractions, including chromatin-bound fractions (*Figure 1—figure supplement 1C*). We also detected a large decrease in bound TBP at *Gapdh* and *Hspa1a* as measured by ChIP-qPCR (*Figure 1—figure supplement 1D*). To further assess the degree of acute depletion of chromatin-bound TBP genome-wide, we performed spike-in controlled Cleavage Under Targets and Tagmentation (CUT&Tag), a high-resolution chromatin profiling technique (*Kaya-Okur et al., 2019*), in two independent replicates after 6 and 16 hr of TBP depletion. After spike-in normalization, CUT&Tag analysis of TBP under control conditions shows high enrichment of TBP at the transcription start sites of genes (TSSs) (*Figure 1C–D*), with high reproducibility across replicates compared to the nonspecific IgG negative control (*Figure 1—figure supplement 1E–G*). After 6 and 16 hr of IAA treatment, we observed a large depletion of bound TBP (*Figure 1C–D*, *Figure 1—figure supplement 1E*), and when quantified at the promoters of genes (herein defined as –250 bp to TSS) relative to control conditions, we observed over ~80% decrease (*Figure 1E*). Pearson correlation confirms the difference between conditions, as control samples correlated more strongly with one another compared to IAA-treated samples (*Figure 1—figure supplement 1G*). Therefore, using orthogonal approaches, we confirm efficient depletion of TBP.

To investigate the immediate effects of TBP depletion in mESCs, we measured Pol II occupancy using CUT&Tag under control and 6 hr of TBP depletion, and normalized the signal using the ChIPseqSpikeInFree method (*Jin et al., 2020*; *Figure 1C and D*) and two independent spike-in controls (*Figure 1—figure supplement 2A*). The normalization methods gave concordant results, hence subsequent analyses for Pol II CUT&Tag was performed using the ChIPseqSpikeInFree method unless stated otherwise. We observed that in control samples, Pol II binds at high levels on the promoters and gene bodies of *Gapdh* and *Actb* (*Figure 1C*, *Figure 1—figure supplement 2B*). After 6 hr of IAA treatment, Pol II occupancy at the *Gapdh* and *Actb* loci remained largely unchanged. We then plotted the genome-wide occupancy of Pol II surrounding the TSS for all genes as heatmaps (*Figure 1D*, *Figure 1—figure supplement 2A*), and also displayed the normalized read counts for Pol II at the promoters of genes with and without IAA treatment as a scatter plot (*Figure 1E*). Overall, we observed no global changes in genome-wide Pol II occupancy after TBP depletion. Prolonged IAA treatment (16 hr) also showed no global effects on Pol II occupancy (*Figure 1—figure supplement 2C*). Indeed, Pearson correlation analysis using 10 kb bins of all Pol II CUT&Tag replicates show high correlation coefficients (0.85–0.97) across both control and IAA-treated samples regardless of treatment times, showing no global differences in Pol II occupancy (*Figure 1—figure supplement 2D*).

To directly measure Pol II activity, we performed NET-seq (native elongating transcript sequencing) analysis in control and TBP-depleted C64 mESCs. NET-seq captures newly transcribed RNA from elongating Pol II at single nucleotide resolution through extensive fractionation (*Mayer et al., 2015*). After normalization with spike-in controls, we detected comparable intronic and exonic signals in *Gapdh* and *Actb* genes for both control and IAA-treated samples (*Figure 1C*, *Figure 1—figure supplement 2B*). We then displayed normalized NET-seq reads as heatmaps in a 2 kb region surrounding the TSS for all genes (*Figure 1D*, *Figure 1—figure supplement 2E*). Consistent with the Pol II CUT&Tag data, we observe no global differences in nascent RNA levels between control and IAA-treated cells. Using k-means clustering, we identified genes with the largest decrease in TBP occupancy after IAA treatment (*Figure 1—figure supplement 3A*). Despite these genes experiencing the largest depletion of TBP upon IAA treatment, Pol II occupancy and nascent RNA levels remained unchanged (*Figure 1—figure supplement 3A*). We also observed no difference in Pol II occupancy and NET-seq signal for promoters containing or lacking a defined TATA-box sequence upon TBP depletion (*Figure 1—figure supplement 3B*). Taken together, our Pol II CUT&Tag and NET-seq data show that acute TBP depletion does not lead to genome-wide disruption of Pol II transcription in mESCs.

We next performed differential gene expression (DGE) analysis using the edgeR Bioconductor package (*Robinson et al., 2010*) to identify genes with statistically significant differences in Pol II occupancy (CUT&Tag) or activity (NET-seq) between control and IAA-treated samples (*Figure 1F*). Consistent with the global average plots and heatmaps, DGE analysis of both Pol II and NET-seq data in control vs. IAA-treated samples show that ~99.9% of genes do not display significant changes upon

TBP depletion (black dots in *Figure 1F*). However, 29 genes showed significant downregulation with approximately one- to fourfold decrease in Pol II occupancy upon IAA treatment (*Supplementary file 1*). Gene browser tracks for Pol II occupancy and NET-seq signal on *Rab26* and *Eif4a2* genes confirm a decrease upon IAA treatment (*Figure 1—figure supplement 3C*). However, prolonged auxin treatment (16 hr) was also analyzed using DGE analysis and only two genes showed a significant downregulation in Pol II occupancy (*Figure 1—figure supplement 2C*), suggesting that some of the identified downregulated genes at 6 hr may be false positives. Therefore, in contrast to the global effects of TBP depletion in yeast cells, these findings show that only a select number of genes may be affected upon TBP depletion in mouse ESCs.

## TBP is required for RNA Pol III transcription of tRNA genes

Previous studies have shown that TBP is required for initiation of the three main eukaryotic RNA polymerases (*Cormack and Struhl, 1992*). Whereas Pol II transcribes protein coding genes, Pol III transcribes all of the tRNAs, the 5S ribosomal RNA, 7SK, and 7SL RNAs, the U6 spliceosomal RNA, and other groups of non-coding RNAs that are critical for cellular functions (*Roeder and Rutter, 1970*; *Weinmann and Roeder, 1974*; *Price and Penman, 1972*; *Zieve et al., 1977*; *Reddy et al., 1987*; *Wolin and Steitz, 1983*; *Martignetti and Brosius, 1995*). Given our findings for Pol II, we next asked if TBP is required for Pol III transcription in mESCs. Treatment of C64 cells with 6 hr of IAA reduced global TBP protein levels, but protein levels of the RPC7 subunit of Pol III do not change, as confirmed by western blot of C64 whole cell lysates (*Figure 2—figure supplement 1A*), suggesting that Pol III protein levels are unaffected by TBP depletion. To investigate whether the chromatin occupancy of Pol III is affected, we performed CUT&Tag analysis of Pol III with and without IAA (*Figure 2A–B*, *Figure 2—figure supplement 1B*), and validated TBP depletion on tRNA genes in these samples (*Figure 2—figure supplement 1D*). Pol III binds at high levels on tRNA genes *LeuCAA* and *GlnCTG* in control samples, and upon TBP depletion, we observed a major decrease in Pol III occupancy at these tRNA genes and across each individual replicate (*Figure 2A*, *Figure 2—figure supplement 1B*). We also detected Pol III CUT&Tag signal on 5S ribosomal RNA, 7SK, and 7SL RNAs (*Figure 2—figure supplement 1C*), but the effects upon TBP depletion is less consistent, likely due to the highly repetitive nature of these genes. We then plotted the Pol III occupancy in a 2 kb window surrounding the TSS for all tRNAs in control and TBP-depleted cells and observed a global decrease in Pol III occupancy across all tRNAs upon TBP depletion (*Figure 2B*, *Figure 2—figure supplement 1D*). We also plotted the normalized read count values for Pol III on the promoters of tRNAs with and without IAA treatment as a scatter plot (*Figure 2C*) and observed that most points fall below the diagonal, confirming a major effect of TBP depletion on Pol III binding at tRNA genes.

NET-seq can also capture the activity of elongating Pol III (*Mayer et al., 2015*). We therefore analyzed NET-seq signals on tRNA genes in control and TBP-depleted mESCs. We detected high signals for tRNA genes *LeuCAA* and *GlnCTG* in control cells, and observed a two- to threefold decrease upon IAA treatment (*Figure 2A*, *Figure 2—figure supplement 1B*). We also detected subtle changes in the NET-seq signal on 5S ribosomal RNA, 7SK, and 7SL RNA genes (*Figure 2—figure supplement 1C*), with the caveat that the highly repetitive nature of the reads may obscure the real effects. We displayed normalized NET-seq reads as heatmaps in a 50 bp region surrounding the TSS for all tRNAs, and as a scatter plot with normalized read counts, and observed a similar decrease of signal for all tRNAs upon TBP depletion (*Figure 2B–C*, *Figure 2—figure supplement 1D*). Therefore, TBP is required for the transcription of tRNAs by Pol III in mESCs, consistent with previous studies in yeast and in vitro systems (*Cormack and Struhl, 1992*; *White et al., 1992*; *Han et al., 2018*; *Wang and Stumph, 1995*). Importantly, the consistent role of TBP in Pol III transcription contrasts strikingly with the TBP-independent Pol II transcription in mESCs.

## TBP is not required for gene activation

Previously, we have shown that depletion of TBP in mESCs specifically during mitosis led to impaired reactivation of Pol II genes as cells enter G1 phase (*Teves et al., 2018*). Based on these observations, we hypothesized that a potential mechanism for TBP-independent Pol II transcription is that TBP may be required to activate silenced genes, but that subsequent transcriptional events would no longer require TBP. To test this hypothesis, we turned to the HS response. A highly conserved protective mechanism to various stressors, the HS response leads to rapid transcriptional induction of HS

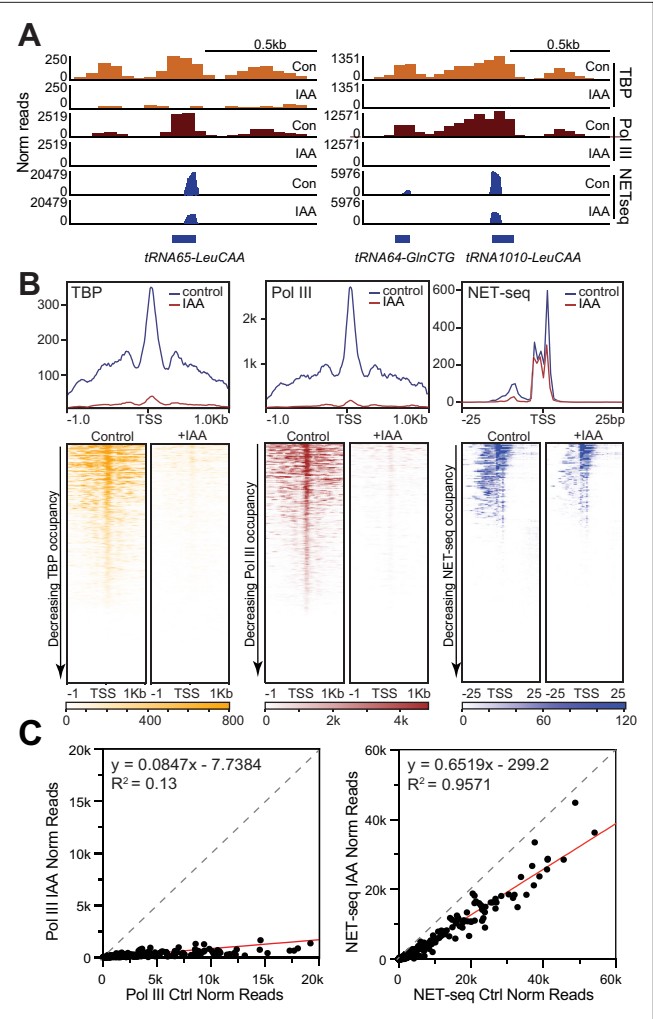

**Figure 2.** TATA-box binding protein (TBP) is required for Pol III transcription of tRNAs in mouse embryonic stem cells (mESCs). (**A**) Gene browser tracks of *tRNA65-LeuCAA* (top) and *tRNA64-GlnCTG/tRNA1010-LeuCAA* (bottom) for cleavage under targets and tagmentation (CUT&Tag) analyses of TBP (orange), Pol III (brown), and NET-seq data (blue) in control or indole-3-acetic acid (IAA)-treated C64 mouse embryonic stem cells (mESCs). (**B**) Average plots (top) and heatmaps arranged by decreasing occupancy (bottom) for TBP CUT&Tag (left), Pol III CUT&Tag (middle), and NET-seq (right) in a 2 kb window (CUT&Tag) or 50 bp window (NET-seq) surrounding the transcription start site (TSS) of all tRNAs. (**C**) Normalized read counts of Pol III CUT&Tag (top) and NET-seq signal (bottom) for control vs. IAA-treated mESCs in either the promoter (–250 bp to TSS) region of each tRNA gene for CUT&Tag or 50 bp window surrounding the TSS of all tRNAs for NET-seq.

The online version of this article includes the following source data and figure supplement(s) for figure 2:

**Figure supplement 1.** Replicate analysis of TATA-box binding protein (TBP) and Pol III cleavage under targets and tagmentation (CUT&Tag).

**Figure supplement 1—source data 1.** Full uncropped membrane scans of immunoblot analyses with α-RPC7 for control and indole-3-acetic acid (IAA)-treated C64 whole cell lysates without and with annotations.

**Figure supplement 1—source data 2.** Full uncropped membrane scans of immunoblot analyses with α-TBP control and indole-3-acetic acid (IAA)-treated C64 whole cell lysates without and with annotations.

**Figure supplement 1—source data 3.** Full uncropped membrane scans of immunoblot analyses with α-Tubulin control and indole-3-acetic acid (IAA)-treated C64 whole cell lysates without and with annotations.

protein genes (*DiDomenico et al., 1982*). We exposed C64 cells to HS at 42°C for 30 min (*Figure 3A*) before collecting cells for Pol II CUT&Tag and NET-seq analyses. Gene browser tracks and heatmaps for CUT&Tag and NET-seq at HS genes show massive increases in occupancy and activity upon HS compared to control samples (*Figure 3B*, *Figure 3—figure supplement 1A–C*). DGE analysis of Pol

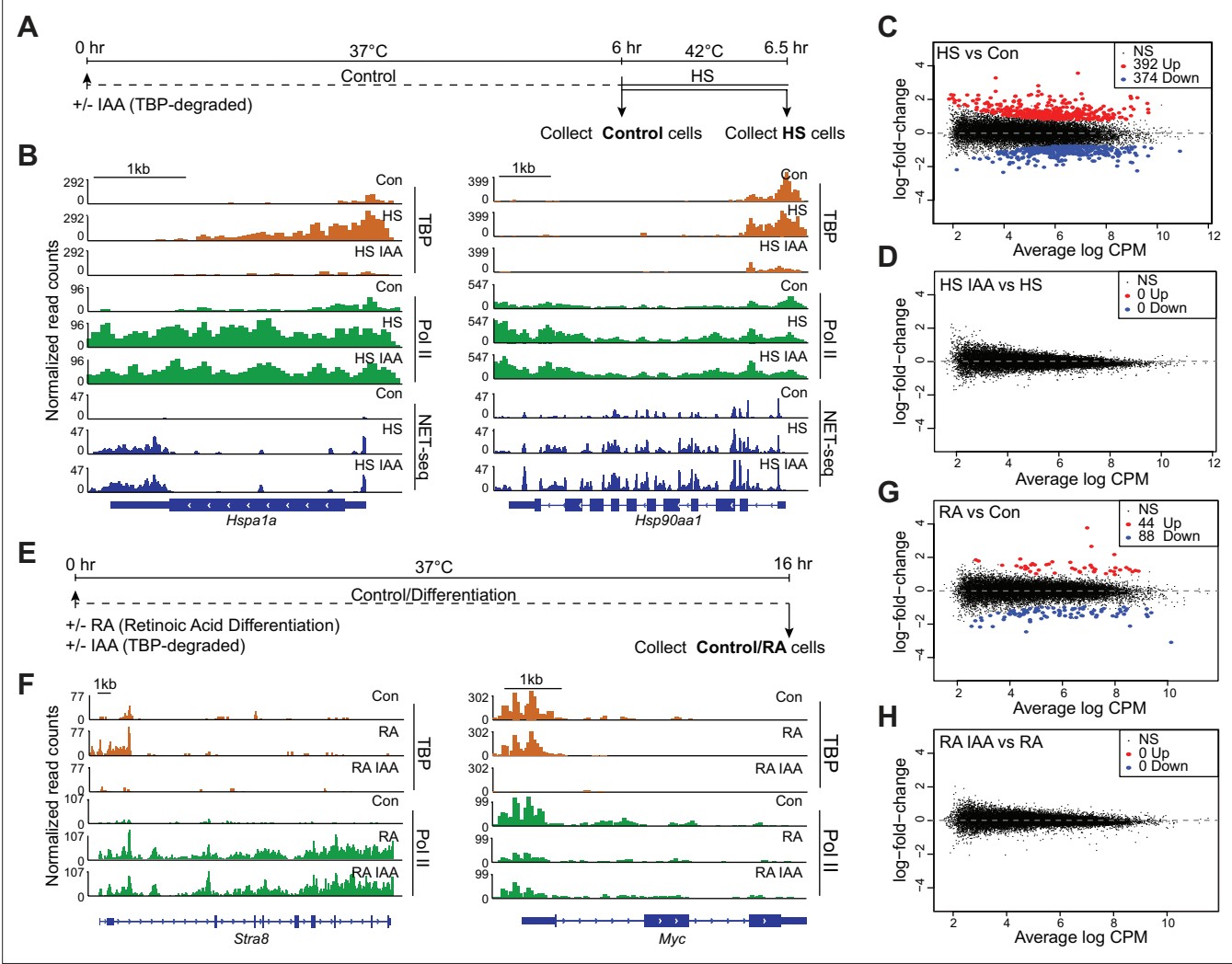

**Figure 3.** TATA-box binding protein (TBP) is dispensable for gene activation by heat shock and retinoic acid differentiation in mouse embryonic stem cells (mESCs). (**A**) Schematic of heat shock treatments for C64 mESCs. After 6 hr of treatment with auxin (indole-3-acetic acid [IAA]) or DMSO, cells were either collected or subjected to heat shock at 42°C for 30 min before collection. (**B**) Gene browser tracks of *Hspa1a* (left) and *Hsp90aa1* (right) for cleavage under targets and tagmentation (CUT&Tag) analyses of TBP (orange), Polymerase II (Pol II) (green), and NET-seq data (blue) in control or IAA-treated C64 mESCs. (**C–D**) Differential gene expression (DGE) analysis of Pol II CUT&Tag in heat shocked vs. control C64 mESCs (**C**) and in heat shocked + IAA-treated vs. heat shocked C64 mESCs (**D**). (**E**) Schematic of retinoic acid (RA) treatment for C64 mESCs. Cells were treated for 16 hr with DMSO or IAA and RA before collection. (**F**) Gene browser tracks of *Stra8* (left) and *Myc* (right) for CUT&Tag analyses of TBP (orange), Pol II (green) in control or IAA-treated C64 mESCs. (**G–H**) DGE analysis of Pol II CUT&Tag in RA-treated vs. control C64 mESCs (**G**) and in RA + IAA-treated vs. RA-treated C64 mESCs (**H**).

The online version of this article includes the following figure supplement(s) for figure 3:

**Figure supplement 1.** Replicate analysis of TATA-box binding protein (TBP) and Polymerase II (Pol II) cleavage under targets and tagmentation (CUT&Tag), and NET-seq data on heat shock (HS)- and retinoic acid (RA)-induced genes.

II CUT&Tag data shows that 392 and 374 genes are significantly upregulated and downregulated, respectively, in HS samples compared to control (***Figure 3C***, ***Supplementary file 2***). Gene ontology (GO) analysis of the upregulated genes showed markers that are typically enriched for HS response, further confirming proper HS induction (***Supplementary file 3***).

To investigate the effects of TBP depletion on the induction of HS genes, we treated C64 cells with IAA for 6 hr followed by HS for 30 min and performed Pol II CUT&Tag and NET-seq analyses (***Figure 3A***). We confirmed TBP depletion following IAA treatment on HS genes using CUT&Tag (***Figure 3—figure supplement 1A–B***). Under HS + IAA conditions, gene browser tracks for Pol II CUT&Tag and NET-seq at HS genes show that these genes are still induced upon TBP depletion

(*Figure 3B*, *Figure 3—figure supplement 1A*). DGE analysis of Pol II CUT&Tag for HS versus HS + IAA samples shows no significant changes in gene expression (*Figure 3D*), indicating that global expression changes induced by HS still occur upon TBP depletion. We also plotted Pol II occupancy for the top 275 upregulated HS genes identified from DGE analysis as heatmaps centered at the TSS (*Figure 3—figure supplement 1C*) and observed no change between HS and HS + IAA conditions. These results suggest that depletion of TBP does not impair Pol II induction of HS genes.

The HS response is a classical gene induction system regulated at the level of paused Pol II elongation (*Core and Adelman, 2019*), and HS genes may retain a scaffold of PIC components following promoter pausing release. Therefore, we also examined the role of TBP in an orthogonal gene activation system through RA-mediated differentiation of C64 mESCs, which leads to the silencing of pluripotency markers and activation of ectoderm-specific genes (*Semrau et al., 2017*). We treated C64 cells with RA and IAA for 16 hr (*Figure 3E*), and performed Pol II CUT&Tag. Gene browser tracks of *Stra8* and *Cdx1* genes show significant increases in Pol II occupancy after RA treatment (*Figure 3F*, *Figure 3—figure supplement 1D*). Additionally, the pluripotency marker *Myc* displays a large decrease in Pol II occupancy under RA conditions (*Figure 3F*). DGE analysis of Pol II CUT&Tag revealed 44 upregulated and 88 downregulated genes, confirming proper RA induction (*Figure 3G*, *Supplementary file 4*).

We then performed Pol II CUT&Tag in TBP-depleted and RA-treated cells (*Figure 3E*), and verified TBP depletion using CUT&Tag analysis (*Figure 3F*, *Figure 3—figure supplement 1D–F*). Gene browser tracks for Pol II CUT&Tag at the *Stra8* gene, a RA-mediated gene, show similar levels of Pol II binding in both RA and RA + IAA conditions (*Figure 3F*). DGE analysis of Pol II CUT&Tag data for RA and RA + IAA samples (*Figure 3H*) also showed no significant differences between the two conditions. Finally, we plotted the Pol II occupancy for the 44 significantly upregulated genes extracted from DGE analysis for RA and RA + IAA treatment as heatmaps centered at the TSS (*Figure 3—figure supplement 1F*). Both RA and RA + IAA conditions display similar levels of Pol II, confirming that depleting TBP does not affect the activation of RA-specific genes. Taken together, these results indicate that TBP is dispensable for both ongoing transcription and gene induction.

## The TBP paralog TRF2 is not required for Pol II transcription

As a second potential mechanism for TBP-independent Pol II transcription, we hypothesized that TBP paralogs may functionally replace TBP. The mouse genome contains two known TBP paralogs, the *Tbpl1* and *Tbpl2* genes, which encode for the proteins TBP-related factor 2 and 3 (TRF2 and TRF3), respectively (*Akhtar and Veenstra, 2011*). These proteins were shown to facilitate transcription in place of or in conjunction with TBP in specific organisms (*Akhtar and Veenstra, 2009*; *Akhtar and Veenstra, 2011*), and binds to active promoters along with TBP and TAF7l in mouse haploid testes cells (*Martianov et al., 2016*). We observed that unlike TRF2, TRF3 is not expressed in mESCs as measured by Pol II CUT&Tag signal and chromatin-associated RNA-seq from a previous study (*Teves et al., 2018*; *Figure 4—figure supplement 1A*), and that TRF2 levels are unaffected by TBP depletion (*Figure 4A*).

To profile TRF2 chromatin binding in mESCs, we overexpressed HA-tagged TRF2 (B8HA mESCs) in a TRF2 knock-out C64 mESCs (B8 mESCs) and confirmed expression via western blot of whole cell lysates (*Figure 4B*). We then performed spike-in normalized CUT&Tag for HA-TRF2 using the HA antibody with and without TBP depletion, and confirmed TBP depletion in B8HA cells through TBP CUT&Tag analysis (*Figure 4C*, *Figure 4—figure supplement 1B*). We found that HA-TRF2 binds to the promoter of *Gapdh,* and that the levels remain relatively unchanged upon TBP depletion (*Figure 4C*, *Figure 4—figure supplement 1B*). The HA signal is also specific, showing no binding in the B8 mESCs (*Figure 4C*). We then displayed the HA-TRF2 CUT&Tag data as a heatmap and average plot for all genes (*Figure 4D*). Similar to TBP, TRF2 binds to promoters of all active genes, and occupancy levels remain similar upon IAA treatment. Furthermore, B8HA HA-TRF2 occupancy is positively correlated with TBP and Pol II levels from the C64 cell line (*Figure 4—figure supplement 1C–D*). Therefore, in contrast to other species, mouse TRF2 has the capacity to bind to the same promoters of active genes as TBP in mESCs.

To test whether TRF2 can functionally replace TBP, we generated a TRF2 knock-out cell line with the entire coding sequence removed (B8 cells). We validated the knock-out with western blot of whole cell lysates and qRT-PCR (*Figure 4B*, *Figure 4—figure supplement 1E–F*). We then performed

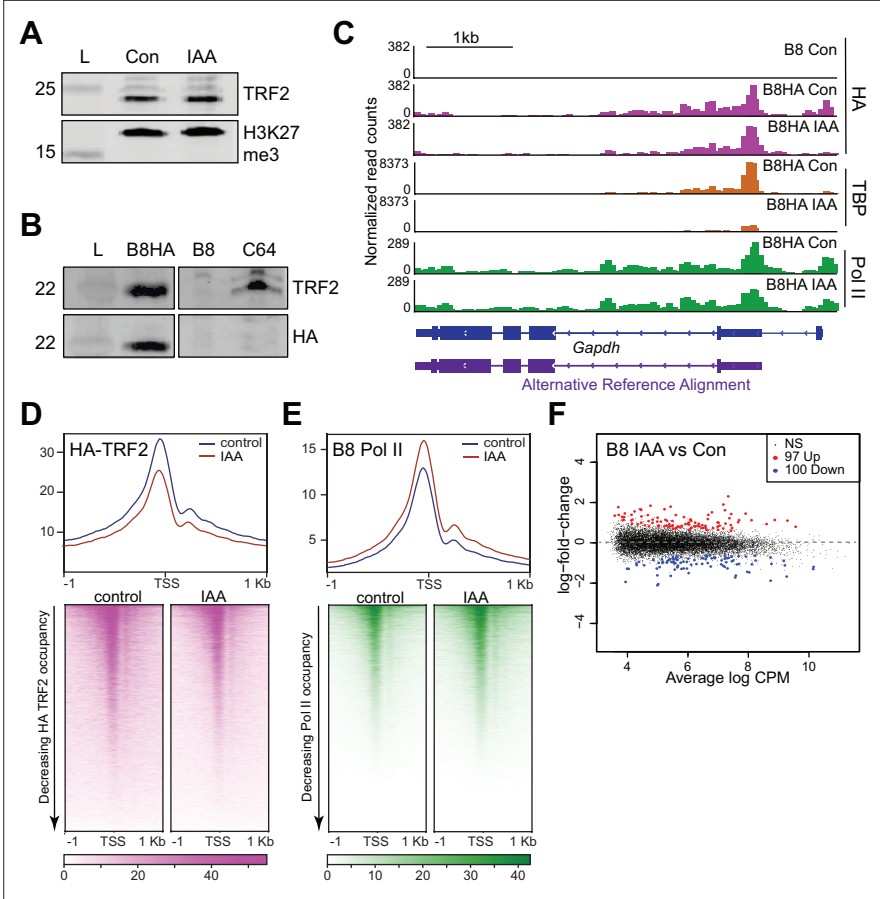

**Figure 4.** TATA-box binding protein (TBP) paralog TBP-related factor 2 (TRF2) is expressed in mouse embryonic stem cells (mESCs), but does not functionally replace TBP. (**A**) Western blot analyses of whole cell lysates for α-TRF2 and α-H3K27me3 in control and indole-3-acetic acid (IAA)-treated C64 mESCs. (**B**) Western blot analyses of whole cell lysates for α-TRF2 and α-HA in the TRF2 knock-out cells overexpressing HA-TRF2 (B8HA), TRF2 knock-out cell line (B8), and C64. α-HA shows the TRF2 band in only the B8HA cell line, indicating proper and specific expression of HA-TRF2. (**C**) Gene browser tracks of *Gapdh* for cleavage under targets and tagmentation (CUT&Tag) analyses of α-HA (magenta), TBP (orange), and Polymerase II (Pol II) (green), in control or IAA-treated B8HA cell line. (**D**) Genome-wide average plot (top) and heatmap (bottom) arranged by decreasing HA-TRF2 occupancy of α-HA CUT&Tag in a 2 kb window surrounding the transcription start site (TSS) of all genes for control and IAA-treated B8HA mESCs. (**E**) Genome-wide average plot (top) and heatmap (bottom) arranged by decreasing Pol II occupancy of α-Pol II CUT&Tag in a 2 kb window surrounding the TSS of all genes for control and IAA-treated B8 mESCs. (**F**) Differential gene expression (DGE) analysis of Pol II CUT&Tag in control vs. IAA-treated B8 mESCs.

The online version of this article includes the following source data and figure supplement(s) for figure 4:

**Source data 1.** Full uncropped membrane scans of immunoblot analyses with α-TRF2 and α-H3K27me3 control and indole-3-acetic acid (IAA)-treated C64 whole cell lysates without and with annotations.

**Source data 2.** Full uncropped membrane scans of immunoblot analyses with α-HA control B8HA, B8, and C64 whole cell lysates without and with annotations.

**Source data 3.** Full uncropped membrane scans of immunoblot analyses with α-TRF2 control B8HA, B8, and C64 whole cell lysates without and with annotations.

**Figure supplement 1.** Characterization of TRF2 knock-out B8 cells.

**Figure supplement 1—source data 1.** Full uncropped membrane scans of immunoblot analyses with α-TRF2 control B8 and C64 whole cell lysates without and with annotations.

qRT-PCR to quantify the intronic and exonic transcripts of *Gapdh* in B8 cells with and without IAA treatment (*Figure 4—figure supplement 1G*). We observed no differences in both intronic and exonic RNA levels for *Gapdh* despite the combination of TBP knockdown and TRF2 knock-out, suggesting that nascent transcription of *Gapdh* is unaffected. To determine the effects on Pol II upon knocking out TRF2, we performed Pol II CUT&Tag on the B8 cells. We then performed DGE analysis of the Pol II CUT&Tag in the TRF2 knock-out B8 cells versus the control C64 mESCs (*Figure 4—figure supplement 1H*). DGE shows only 13 upregulated genes and 8 downregulated genes, suggesting that knock-out of TRF2 has no global consequences. To further investigate if TRF2 and TBP are functionally redundant in mESCs, we performed CUT&Tag for TBP and Pol II on TRF2 knock-out B8 cells with and without TBP depletion (*Figure 4E*, *Figure 4—figure supplement 1I–J*). We confirmed proper depletion of TBP in the B8 cell line (*Figure 4—figure supplement 1I*), and Pearson correlation analysis using 10 kb bins of Pol II CUT&Tag shows good concordance between replicates (*Figure 4—figure supplement 1K*). DGE analysis shows that 100 and 97 genes were downregulated and upregulated, respectively, upon TBP depletion in the B8 cells, confirming that Pol II occupancy is not affected globally upon TBP depletion in the TRF2 knock-out cells (*Figure 4F*). Additionally, we performed a cell viability assay under prolonged IAA treatment when either TBP, TRF2, or both are absent (*Figure 4—figure supplement 1L*). Both B8 and C64 cells displayed a similar death curve when treated with IAA, suggesting that the combinatorial knock-out of TRF2 with TBP depletion does not accelerate cell death. We conclude that TRF2 is not functionally redundant to TBP and that Pol II transcription may be TBP family-independent in mESCs.

## The TFIID complex forms upon TBP depletion

A third potential mechanism for TBP-independent Pol II transcription is that TAF-containing TFIID complexes could function when TBP is depleted. To examine the composition of TFIID upon TBP depletion, we performed co-immunoprecipitation followed by mass spectrometry (IP-MS) with α-TAF4 in control and IAA-treated cells (*Figure 5—figure supplement 1A–B*). TAF4 is a member of the core TFIID subcomplex (*Wright et al., 2006*), and is also present in other TAF-containing complexes like TFTC/SAGA (*Wieczorek et al., 1998*). We confirmed TBP depletion, TAF4 expression, and TAF4 IP in control and IAA-treated cells by western blot analysis (*Figure 5A*). From our TAF4 IP-MS analysis, we observed an 80% decrease of co-precipitated TBP in IAA-treated cells compared to control cells (*Figure 5B*), although we observed a greater decrease by western blot analyses of whole cell lysates (*Figure 5A*). In the TAF4 IP-MS of control cells, we identified peptides for all of the TAF subunits of the TFIID complex, as well as some peptides for TAF paralogs like TAF4b and TAF9b. Importantly, in the IAA-treated cells, we detected similar levels of enrichment for the identified TAF subunits (*Figure 5C*, *Figure 5—figure supplement 1A–B and D*). Statistical analyses of replicates show no significant differences in the enrichment levels of TFIID members by TAF4 IP-MS in control vs. IAA-treated cells. These results suggest that the full TFIID complex can form even when TBP is depleted. Furthermore, the nuclear TFIID complex does not dissociate into smaller subcomplexes when TBP is depleted, as gel filtration analysis from control and TBP-depleted nuclear lysates showed no change in TAF4 elution patterns. TAF4 is only detected at the earliest fractions, indicative of the high molecular weight of the full TFIID complex, in both control and TBP-depleted nuclear lysates (*Figure 5—figure supplement 1C*). Although the gel filtration only provides information on the nuclear complex, we observed no changes in relative stoichiometries of the TAF subunits, as measured by the TAF4 IP-MS (*Figure 5—figure supplement 1D*), suggesting that TBP depletion has no effect on any TAF4 subcomplexes formed in the cytoplasm. Taken together, these results indicate that, even under TBP-depleted conditions, the full TFIID complex can form, and that any subcomplexes containing TAF4 are unaffected.

## Effects of TBP depletion on DNA binding of TAF subunits and TFIIA

Next, we examined the DNA-binding profiles of certain TAF subunits. TAF1 is the largest subunit of TFIID, interacts with TBP through an N-terminal TBP-binding sequence, and contains multiple DNA-binding domains (*Curran et al., 2018*). We performed TAF1 CUT&Tag in control and TBP-depleted C64 mESCs. TAF1 binds at the promoter regions of *Actb* and *Gapdh* at similar levels in control and IAA-treated cells (*Figure 6A*, *Figure 6—figure supplement 1A*). We then plotted the genome-wide occupancy of TAF1 in a 2 kb window surrounding the TSS for all genes as heatmaps (*Figure 6B*, *Figure 6—figure supplement 1B*), and displayed the normalized read counts for TAF1

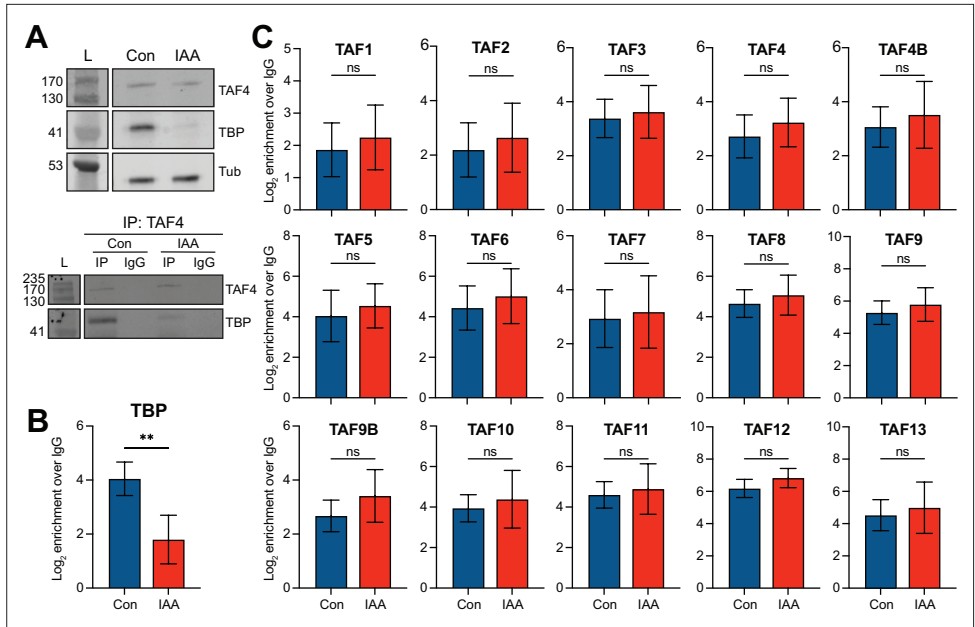

**Figure 5.** The TFIID complex forms upon TATA-box binding protein (TBP) depletion. (**A**) Top: Western blot analyses for α-TAF4, α-TBP, and α-Tubulin of control and indole-3-acetic acid (IAA)-treated C64 whole cell extracts used in the TAF4 IP-MS analyses. Input represents 5% of the protein samples used in the TAF4 IPs. Bottom: Western blot analyses for α-TAF4 and α-TBP of TAF4 IPs from control and IAA-treated C64 whole cell extracts. (**B–C**) Boxplots showing abundance of TBP (**B**) and the TBP-associated factors (TAFs) (**C**) from α-TAF4 IP-MS analyses of control (blue) and IAA-treated (red) C64 mouse embryonic stem cells (mESCs). Protein levels were normalized to levels in α-IgG pull-downs from the same cells. Two asterisks represent p≤0.01 using standard two-tailed t-test. Error bars represent standard deviation of n=4.

The online version of this article includes the following source data and figure supplement(s) for figure 5:

**Source data 1.** Full uncropped membrane scans of immunoblot analyses with α-TAF4 for control and indole-3-acetic acid (IAA)-treated C64 whole cell extracts without and with annotations.

**Source data 2.** Full uncropped membrane scans of immunoblot analyses with α-TBP for control and indole-3-acetic acid (IAA)-treated C64 whole cell extracts without and with annotations.

**Source data 3.** Full uncropped membrane scans of immunoblot analyses with α-Tubulin for control and indole-3-acetic acid (IAA)-treated C64 whole cell extracts without and with annotations.

**Source data 4.** Full uncropped membrane scans of immunoblot analyses with α-TAF4 for TAF4 IPs from control and indole-3-acetic acid (IAA)-treated C64 whole cell lysates without and with annotations.

**Source data 5.** Full uncropped membrane scans of immunoblot analyses with α-TBP for TAF4 IPs from control and indole-3-acetic acid (IAA)-treated C64 whole cell lysates without and with annotations.

**Figure supplement 1.** TAF4 IP-MS analyses in control and TATA-box binding protein (TBP)-depleted cells.

**Figure supplement 1—source data 1.** Full uncropped membrane scans of immunoblot analyses with α-TAF4 for gel filtration fractions 1–24 from control C64 cells without and with annotations.

**Figure supplement 1—source data 2.** Full uncropped membrane scans of immunoblot analyses with α-TAF4 for gel filtration fractions 25–47 from control C64 cells without and with annotations.

**Figure supplement 1—source data 3.** Full uncropped membrane scans of immunoblot analyses with α-TBP for gel filtration fractions 1–24 from control C64 cells without and with annotations.

**Figure supplement 1—source data 4.** Full uncropped membrane scans of immunoblot analyses with α-TBP for gel filtration fractions 25–47 from control C64 cells without and with annotations.

**Figure supplement 1—source data 5.** Full uncropped membrane scans of immunoblot analyses with α-TAF4 for gel filtration fractions 1–24 from indole-3-acetic acid (IAA)-treated C64 cells without and with annotations.

**Figure supplement 1—source data 6.** Full uncropped membrane scans of immunoblot analyses with α-TAF4 for gel filtration fractions 25–47 from indole-3-acetic acid (IAA)-treated C64 cells without and with annotations.

**Figure supplement 1—source data 7.** Full uncropped membrane scans of immunoblot analyses with α-TBP for gel filtration fractions 1–24 from indole-3-acetic acid (IAA)-treated C64 cells without and with annotations.

*Figure 5 continued*

**Figure supplement 1—source data 8.** Full uncropped membrane scans of immunoblot analyses with α-TBP for gel filtration fractions 25–47 from indole-3-acetic acid (IAA)-treated C64 cells without and with annotations.

on the promoter of all genes with and without IAA treatment as a scatter plot (*Figure 6C*). From the heatmaps and scatter plots, we observed no major changes in TAF1 occupancy after TBP depletion. Pearson correlation analysis using 10 kb bins or surrounding the TSS (-/+1000 bp) shows that the TAF1 control and IAA samples correlate with one another (*Figure 6—figure supplement 1C*). Furthermore, k-means clustering for changes in TBP binding upon IAA treatment shows that TAF1 binding does not change for genes that experience the largest depletion of TBP (*Figure 6—figure supplement 3A*, cluster 3). Scatter plot analysis of TAF1 signal at each promoter relative to Pol II levels show that genes do not decrease in either TAF1 or Pol II occupancy upon TBP depletion (*Figure 6—figure supplement 3B*). Therefore, TAF1 binding is unaffected by TBP depletion.

We next analyzed TAF4 occupancy by performing CUT&Tag in control and TBP-depleted C64 mESCs. We observed that TAF4 binds on the promoters of *Gapdh* and *Actb* in control cells (*Figure 6A*, *Figure 6—figure supplement 1A*); however, after TBP depletion, we observed a moderate decrease of TAF4 occupancy at these loci. We plotted TAF4 occupancy in a 2 kb window surrounding the TSS of all genes in control and TBP-depleted cells as heatmaps and observed a similar decrease in TAF4 occupancy across all genes upon TBP depletion (*Figure 6B*, *Figure 6—figure supplement 1D*). We also plotted the normalized TAF4 read counts on the promoter of all genes with and without IAA treatment as a scatter plot (*Figure 6C*) and observed that most points fall below the diagonal with about 65% decrease in signal. Pearson correlation analysis using 10 kb bins or surrounding the TSS (-/+1000 bp) shows that the TAF4 control samples correlate more strongly with one another than with the IAA-treated, confirming that TBP depletion does indeed affect TAF4 occupancy (*Figure 6—figure supplement 1D*). Furthermore, when we performed k-means clustering of the TAF4 CUT&Tag data, we observed that genes with the largest decrease in TBP occupancy after IAA treatment also experienced the greatest decrease in TAF4 occupancy (*Figure 6—figure supplement 3A*). Scatter plot analysis of TAF4 relative to Pol II shows that genes that experience a decrease in TAF4 occupancy upon TBP depletion do not decrease in Pol II occupancy (*Figure 6—figure supplement 3B*). Therefore, unlike TAF1, TAF4 binding is affected by TBP depletion.

Previous studies have shown that TBP mediates the interaction between TFIID and TFIIA (*Kraemer et al., 2001*). Indeed, in the TAF4 IP-MS, we detected high levels of TFIIA-γ peptides under control conditions that significantly decreased in the IAA-treated conditions (*Figure 6—figure supplement 2A*), confirming that TBP mediates this interaction. The decrease in interaction is not due to altered protein levels, as measured by whole cell lysate western blot (*Figure 6—figure supplement 2A*). To determine the effects of TBP depletion on TFIIA binding to promoters, we performed CUT&Tag using an antibody that detects the second subunit of TFIIA, with high concordance between replicates (*Figure 6—figure supplement 2B–D*). We observe that TFIIA binds to the promoters of *Gapdh* and *Actb* in control conditions, and that this occupancy is decreased upon IAA treatment (*Figure 6A*, *Figure 6—figure supplement 2B*). We then plotted TFIIA occupancy in a 2 kb window surrounding the TSS of all genes in control and IAA-treated cells as a heatmap and observed a global decrease in occupancy upon TBP depletion (*Figure 6B*, *Figure 6—figure supplement 2C*). Scatter plot analysis of normalized reads on the promoters of all genes in control versus TBP-depleted cells show about 60% decrease in overall signal (*Figure 6C*). Pearson correlation analysis using 10 kb bins or surrounding the TSS (-/+1000 bp) show that TFIIA IAA-treated samples correlate more closely with one another than with the control samples and vice versa (*Figure 6—figure supplement 2D*). Additionally, k-means clustering of the TFIIA CUT&Tag data shows a similar trend to that of TAF4, where genes that experience the greatest decrease in TBP depletion (cluster 3) also show the largest reduction in TFIIA signal (*Figure 6—figure supplement 3A*). Scatter plot analysis of TFIIA signal relative to Pol II shows that although the occupancy of TFIIA decreases, Pol II occupancy is unaffected (*Figure 6—figure supplement 3C*). A similar analysis between TFIIA and TAF4 (*Figure 6—figure supplement 3C*) shows that genes experiencing a decrease in TAF4 occupancy also decrease in TFIIA binding.

Lastly, we plotted TBP, Pol II, TAF4, TAF1, TFIIA CUT&Tag, and our NET-seq data as heatmaps (top) and average plots (bottom) arranged by decreasing TBP occupancy and observe that TBP occupancy correlates strongly with transcription levels (Pol II and NET-seq data) and with the other GTFs in

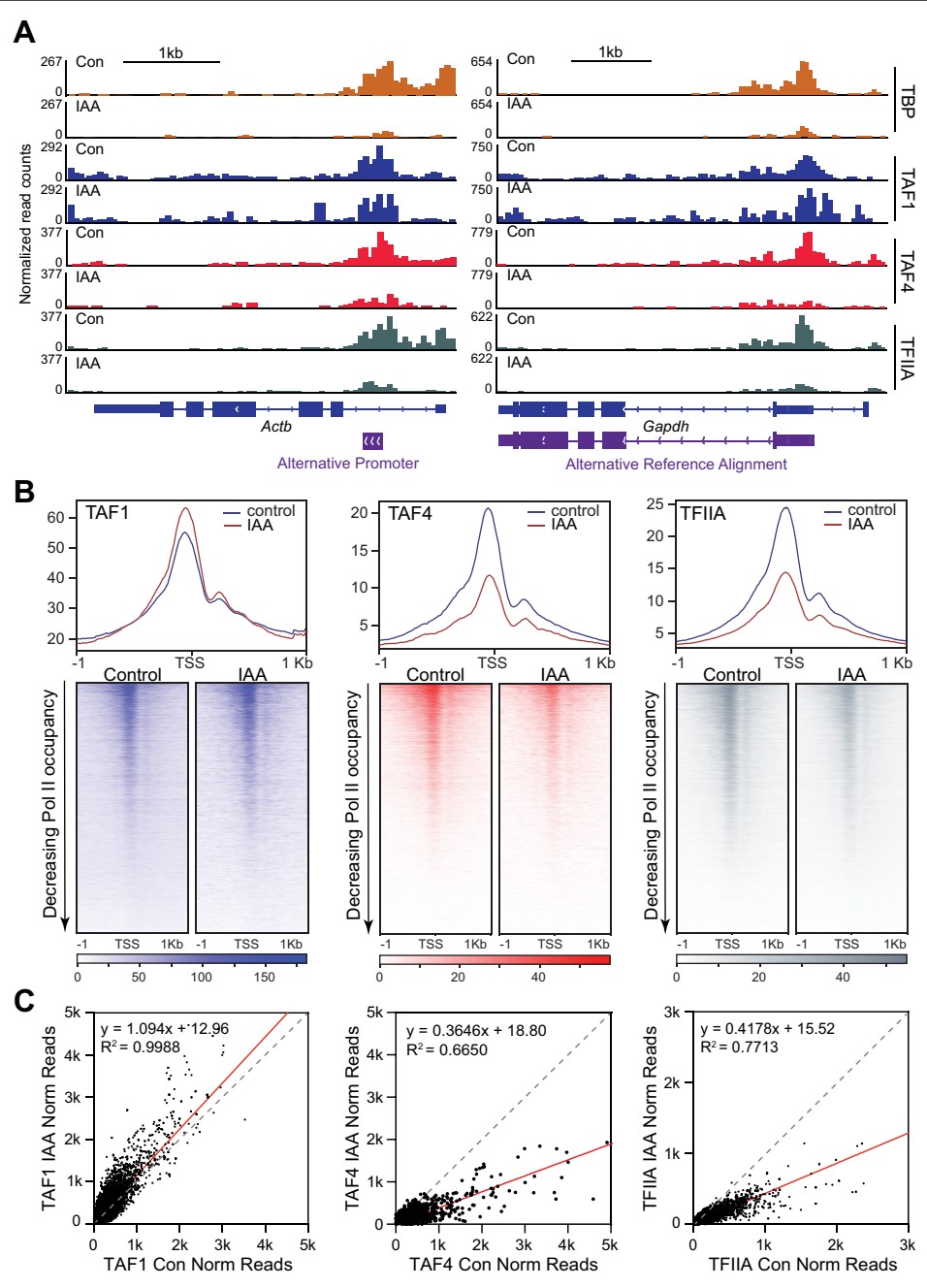

**Figure 6.** Effects of TATA-box binding protein (TBP) depletion on the DNA binding of TAF1, TAF4, and TFIIA. (**A**) Gene browser tracks of *Actb* (top) and *Gapdh* (bottom) for cleavage under targets and tagmentation (CUT&Tag) analyses of TAF1 (blue), TAF4 (red), and TFIIA (gray). TBP signal (orange) from *Figure 1* is replotted for comparison. (**B**) Genome-wide average plots (top) and heatmaps (bottom) arranged by decreasing Polymerase II (Pol II) occupancy for TAF1 (left), TAF4 (middle), and TFIIA (right) CUT&Tag in a 2 kb window surrounding the transcription start site (TSS) of all genes for control and indole-3-acetic acid (IAA)-treated C64 mouse embryonic stem cells (mESCs). (**C**) Normalized read counts of TAF1 (left), TAF4 (middle), and TFIIA (right) CUT&Tag for control vs. IAA-treated C64 mESCs at the promoter (–250 bp to the TSS) region for all genes.

The online version of this article includes the following source data and figure supplement(s) for figure 6:

**Figure supplement 1.** Replicate analysis of Taf1 and Taf4 cleavage under targets and tagmentation (CUT&Tag), and NET-seq data.

**Figure supplement 2.** Replicate analysis of TFIIA cleavage under targets and tagmentation (CUT&Tag).

*Figure 6 continued on next page*

*Figure 6 continued*

**Figure supplement 2—source data 1.** Full uncropped membrane scans of immunoblot analyses with α-TBP for control and indole-3-acetic acid (IAA)-treated C64 whole cell lysates without and with annotations.

**Figure supplement 2—source data 2.** Full uncropped membrane scans of immunoblot analyses with α-TFIIA-γ for control and indole-3-acetic acid (IAA)-treated C64 whole cell lysates without and with annotations.

**Figure supplement 2—source data 3.** Full uncropped membrane scans of immunoblot analyses with α-Tubulin for control and indole-3-acetic acid (IAA)-treated C64 whole cell lysates without and with annotations.

**Figure supplement 3.** Comparisons between Polymerase II (Pol II) TAF1, TAF4, and TFIIA.

---

control conditions, as expected (*Figure 6—figure supplement 3D*). Furthermore, highly bound TBP genes show the greatest reduction in TFIIA and TAF4 occupancy when TBP is depleted, but experience no differences in Pol II and TAF1. Taken together, our data suggest that TBP depletion disrupts TFIID-TFIIA association and negatively affects TFIIA and TAF4 binding to promoters. However, despite these effects on TAF4 and TFIIA, the Pol II machinery remains sufficiently robust in supporting Pol II transcription.

## Discussion

In this study, we took advantage of a drug-inducible degradation system to deplete TBP rapidly and examined the mechanistic role of TBP in Pol II transcription in mESCs. We found that TBP is largely not required for ongoing Pol II transcription. TBP is, however, required for transcription of tRNA genes by Pol III. We tested several possible mechanisms for TBP-independent Pol II transcription, and found that TBP is not required for gene induction, and that the TBP paralog TRF2 does not functionally replace TBP for Pol II transcription when TBP is depleted. Additionally, the TFIID complex still forms even when TBP is acutely depleted, but that its association with TFIIA is significantly decreased. Although TAF1 binding was unaffected, TAF4 and TFIIA occupancy levels decreased upon TBP depletion. Despite these changes, Pol II activity remains unaffected, suggesting a robustness in the Pol II machinery that compensates for TBP depletion.

Previous studies have examined the role of individual subunits of the TFIID complex in mammalian systems. For instance, knockdown of the TAF5 subunit in human ESCs led to reduced expression of the pluripotency genes (*Pijnappel et al., 2013*), and in mESCs, acute degradation of TAF12 led to a global decrease in gene expression (*Sun et al., 2021*). These studies are consistent with a more prominent role for TFIID. However, a major limitation inherent to the mAID system is the incomplete degradation of the protein, resulting in ~10–20% remaining TBP protein. A previous study has quantified the absolute number of TBP molecules in mESCs to 99,111±29,125 molecules (*Cattoglio et al., 2019*), suggesting ~10,000–20,000 remaining TBP molecules after depletion. From the Pol II CUT&Tag and NET-seq data, ~70% of Refseq TSSs have detectable signals, which results in ~27,185 active genes (from 38,837 total). Based on these numbers, even the highest estimated end of remaining TBP is insufficient to account for one molecule per active TSS. These numbers do not consider Pol II expression that occurs at enhancers or the ribosomal genes transcribed by Pol I. Furthermore, the TBP requirement for Pol III activity in mESCs provides confirmation that the acute TBP depletion system in mESCs has the expected consequences, and emphasizes the lack of effect on Pol II transcription when TBP is depleted. However, these results do not provide information on the functional role of TBP when recruited to Pol II promoters. One possibility is that Pol II transcription has evolved to have multiple redundancies such that removal of one factor (TBP) may not fully disrupt the whole system. Another possibility is that the transient nature of the embryonic cell state, with its unique chromatin landscape, enables Pol II transcription without TBP. Regardless, at the level of TBP depletion that we observe, we conclude that in mESCs Pol II transcription can occur independently of TBP.

We initially hypothesized that the TBP paralog TRF2 could functionally replace TBP in our depletion system. TRF2 is a more distant TBP paralog compared to the other paralogs, sharing only ~40% amino acid identity with the core domain of TBP (*Dantonel et al., 1999*). In *Drosophila*, TRF2 fails to bind to DNA that contains the canonical TATA box. Instead, TRF2 from various species such as *Drosophila*, worms, frogs, fish, and flies has been proposed to activate transcription for TATA-less promoters (*Akhtar and Veenstra, 2011*; *Isogai et al., 2007*). Previous studies have also shown that mouse TRF2 binds to active promoters in haploid testis cells along with TBP and TAF7l (*Martianov*

*et al., 2016*), but the germline may have unique regulatory mechanisms. Our CUT&Tag results show that TRF2 binds to all active promoters in mESCs, and its level of binding correlates strongly to both TBP and Pol II (*Figure 4—figure supplement 1C–D*). One possible explanation for species-specific differences in TRF2 functions is the variability in the TRF2 core domain across species. Unlike TBP, whose core domain is over 80% conserved across species, the core domain of TRF2 only shares 40–45% amino acid conservation between metazoans (*Akhtar and Veenstra, 2011*; *Dantonel et al., 2000*). Although TRF2 has been historically seen to govern specific subsets of genes distinct from TBP, our results show that TRF2 in mESCs may have the potential to substitute for TBP during transcription initiation for mESCs Pol II-transcribed genes. It was thus surprising that depletion of TBP in the TRF2 knock-out mESCs had no global effect on Pol II transcription (*Figure 4E*, *Figure 4—figure supplement 1E*), suggesting that Pol II transcription in mESCs may not only be TBP-independent, but TBP family-independent.

We have shown that the full TFIID complex can form upon TBP depletion and that TFIID does not dissociate into smaller subcomplexes when TBP is depleted, suggesting a resiliency in the TFIID complex. Several studies point to a modular assembly for the full complex, which may allow for such resiliency. For instance, human TAF8 and TAF10 form as a heterodimer in the cytosol (*Kamenova et al., 2019*), with TAF2 joining the subcomplex to promote the nuclear import and incorporation into human TFIID (*Kamenova et al., 2019*; *Soutoglou et al., 2005*; *Trowitzsch et al., 2015*). Similarly, human TAF5, TAF6, and TAF9 have also been shown to form in the cytosol as a subcomplex to promote the incorporation of TAF9 into TFIID (*Antonova et al., 2018*). It is possible that the modular nature of the assembly of TFIID allows for a robust structure even when TBP molecules are severely limiting. Furthermore, TAF7 has been shown to dissociate from the rest of TFIID after transcription initiation and acts as a regulator of RNA translation by binding onto RNAs and delivering them to polysomes for translation in the cytosol (*Gegonne et al., 2006*; *Cheng et al., 2021*). This finding suggests a more dynamic TFIID complex with subunits associating and dissociating in flux, which may allow for built-in redundancy and robustness when a subunit is missing. Such a dynamic TFIID complex would also be consistent with the different effect on TAF1 and TAF4 DNA-binding profiles when TBP is depleted, especially if the absence of TBP leads to changes in TAF-DNA residence times. In contrast, TBP in Pol III initiation forms the TFIIIB complex with only two other factors: Brf2 and Bdp1 (*Gouge et al., 2017*). It is intriguing to speculate that the greater number of TAFs in TFIID (13 or more) might allow for more redundancy than the two other subunits of TFIIIB, providing a potential mechanism to the observed requirement for TBP in Pol III but not for Pol II transcription.

Previous studies have identified the presence of other non-canonical initiation complexes, suggesting a high level of robustness in the machinery to accommodate Pol II transcription. For example, TAF paralogs are often expressed and used in a cell type-specific manner (*Freiman, 2009*), including TAF4b in oocytes and TAF9b in motor neuron differentiation (*Herrera et al., 2014*; *Liu et al., 2011*; *Freiman et al., 2001*; *Pointud et al., 2003*). In *Xenopus* embryos, knockdown of TBP and the two paralogs showed recruitment of Pol II to the promoter and active transcription for a subset of genes (*Gazdag et al., 2016*). In yeast, transcription activation can occur in the absence of TFIIA, and in vitro TFIIA was shown not to be needed for basal transcription (*Chou et al., 1999*; *Sayre et al., 1992*). Additionally, gel mobility-shift analysis and fractionated cross-substitution experiments using *Drosophila* and HeLa systems showed that TFIIA was not required for complexes to be transcriptionally competent (*Tyree et al., 1993*; *Wampler et al., 1990*). Our results provide additional evidence on the robustness of the Pol II machinery. Despite the changes in DNA binding of TAF4 and TFIIA when TBP is depleted, Pol II transcription remains unaffected. Perhaps in the same way that TFIID could behave as a dynamic complex, the entire PIC may also function as a dynamic hub instead of acting as a monolithic complex. Previous studies using single molecule imaging and tracking provide support for this concept. For example, the average residence time of TBP is on the order of minutes whereas TFIIB is on the order of seconds (*Teves et al., 2018*; *Zhang et al., 2016*). Such a dynamic system may provide sufficient robustness to facilitate Pol II transcription when TBP is depleted.

# Materials and methods

## Key resources table

| Reagent type (species) or resource | Designation | Source or reference | Identifiers | Additional information |
|---|---|---|---|---|
| Cell line (*Mus musculus*, male) | JM8.N4 mouse ES cells | KOMP repository | RRID: CVCL_J962 | Parental cell line used for all genetic manipulations |
| Cell line (*Mus musculus*, male) | mAID-TBP KI C94 | https://doi.org/10.7554/eLife.35621 | C94 | Endogenous knock-in of the minimal auxin inducible degron to N-terminus of TBP in JM8.N4 cells |
| Cell line (*Mus musculus*, male) | Halo-Pol II C64 | https://doi.org/10.7554/eLife.35621 | C64 | Endogenous knock-in of the HaloTag to C-terminus of Rbp1, largest subunit of Pol II, in mAID-TBP C94 cells |
| Cell line (*Mus musculus*, male) | B8 | This paper | | Endogenous knock-out of the *Tbpl1* gene from Halo-Pol II C64 cells – see Materials and Methods |
| Cell line (*Mus musculus*, male) | B8HA | This paper | | Overexpressed HA-tagged TRF2 in B8 cells – see Materials and Methods |
| Cell line (*Drosophila melanogaster*) | S2 | Gift from Dr. Eric Jan | | Spike-in of 20% (by cell number) in CUT&Tag and qRT-PCR |
| Transfected construct (*Mus musculus*) | pTRF2-HA | This paper | | HA tagged TRF2 cDNA expressing plasmid – see Materials and Methods |
| Transfected construct (*Mus musculus*) | pCas9-mCherry-TRF2-1 | This paper; backbone: cat #64324 (Addgene) | | Targets *Tbpl1* for knockout - see Materials and Methods |
| Transfected construct (*Mus musculus*) | pCas9-mCherry-TRF2-2 | This paper; backbone: cat #64324 (Addgene) | | Targets *Tbpl1* for knockout - see Materials and Methods |
| Antibody | α-TBP (Mouse, monoclonal) | Abcam | cat:ab51841 | Western blot (1:3000) ChIP (5 µg) CUT&Tag (1 µg) IF (1:100) |
| Antibody | α-Pol II (Rpb1 CTD) | Cell Signaling Technologies | cat:D1G3K | CUT&Tag (1 µg) |
| Antibody | α-H3K27me3 (Rabbit, monoclonal) | Cell Signaling Technologies | cat:C36B11 | Western blot (1:7000) CUT&Tag (1 µg) |
| Antibody | α-HA (Rabbit, polyclonal) | Epicypher | cat:13–2010 | Western blot (1:1000) CUT&Tag (1 µg) |
| Antibody | α-Tubulin (Rabbit, polyclonal) | Abcam | cat:ab6046 | Western blot (1:7000) |
| Antibody | α-Pol III RPC32 (mouse monoclonal) | Santa Cruz Biotechnology | cat:sc-48365 | Western blot (1:2000) |
| Antibody | α-TAF4 (mouse monoclonal) | Santa Cruz Biotechnology | cat:sc-136093 | Western blot (1:2000) CUT&Tag (1 µg) Co-IP (3 µg) |
| Antibody | α-H3.3 (mouse monoclonal) | Abnova | cat:H00003021-M01 | Western blot (1:5000) |
| Antibody | α-TRF2 (mouse monoclonal) | Gift from Dr. László Tora | | Western blot (1:3000) |
| Antibody | α-TFIIA-γ (mouse monoclonal) | Santa Cruz Biotechnology | cat:sc-374483 | Western blot (1:3000) |
| Antibody | α-TFIIA-γ (rabbit polyclonal) | Santa Cruz Biotechnology | cat:sc-25365 | CUT&Tag (1 µg) |
| Antibody | IRDye 800CW α-Mouse IgG (Goat polyclonal) | Licor | cat:926–32210 | Western blot (1:20000) |

*Continued on next page*

*Continued*

| Reagent type (species) or resource | Designation | Source or reference | Identifiers | Additional information |
|---|---|---|---|---|
| Antibody | IRDye 800CW α-Rabbit IgG (Goat polyclonal) | Licor | cat:925–32211 | Western blot (1:20000) |
| Antibody | IRDye 680RD α-Mouse IgG (Goat polyclonal) | Licor | cat:926–68070 | Western blot (1:20000) |
| Antibody | IRDye 680RD α-Rabbit IgG (Goat polyclonal) | Licor | cat:926–68701 | Western blot (1:20000) |
| Antibody | α-Mouse IgG (Rabbit, polyclonal) | Abcam | cat:ab46540 | ChIP (5 µg) CUT&Tag (1:100) |
| Antibody | α-TAF1 (Rabbit, monoclonal) | Cell Signaling Technologies | cat:D6J8B | CUT&Tag (1 µg) |
| Antibody | α-Rabbit IgG (Guinea Pig, polyclonal) | Antibodies-online | cat:ABIN101961 | CUT&Tag (1:100) |
| Antibody | Alexa Fluor 594 α-Mouse IgG (Goat, polyclonal) | Invitrogen | cat:A-11005 | IF (1:100) |
| Antibody | α-Mouse IgG (mouse, monoclonal) | Santa Cruz Biotechnology | cat:sc-2025 | Co-IP (3 µg) |
| Sequence-based reagent | Biotinylated oligonucleotides for NET-seq libraries | This paper | | See Materials and Methods and *Supplementary file 5* for full list |
| Sequence-based reagent | qRT-PCR oligonucleotides | This paper | | See *Supplementary file 6* for full list |
| Sequence-based reagent | ChIP-qPCR oligonucleotides | This paper | | See *Supplementary file 7* for full list |
| Peptide, recombinant protein | pA-Tn5 | Gift from Dr. Steven Henikoff | | Diluted to 1 x in each reaction |
| Peptide, recombinant protein | pAG-Tn5 | Epicypher | cat:15–1017 | Diluted to 1 x in each reaction |

## Cell lines

For all cell lines used in this study, the parental line is JM8.N4 mouse ES cells, purchased from KOMP repository, RRID: CVCL_J962. C64 is a CRISPR-Cas9 genetically modified JM8.N4 cell line containing mAID-TBP knock-in obtained as previously described (*Teves et al., 2018*). B8 is a CRISPR-Cas9 genetically modified C64 cell line with a *Tbpl1* complete gene knock-out. The cell lines have been authenticated by STR profiling, and tested negative for mycoplasma contamination.

## Cell culture

ES cells were cultured on 0.1% gelatin-coated plates in ESC media KnockOut DMEM (Corning) with 15% FBS (HyClone), 0.1 mM MEM non-essential amino acids (Gibco), 2 mM GlutaMAX (Gibco), 0.1 mM 2-mercaptoethanol (Sigma-Aldrich) and 1000 units/ml of ESGRO (Chem-icon). ES cells were fed daily, cultured at 37°C in a 5% $CO_2$ incubator, and passaged every 2 days by trypsinization. For endogenously tagged mAID-TBP C64 and B8 cells, TBP degradation was performed by addition of IAA at 500 µM final concentration to a confluent plate of cells for 6 or 16 hr. HS was performed at 42°C in a 5% $CO_2$ incubator for 30 min. For HS and IAA treatment, cells were first treated with 6 hr of auxin followed by an additional 30 min before being collected. RA (Sigma-Aldrich R2625-100MG) treatment was performed at 37°C in a 5% $CO_2$ incubator at 0.25 µM for 16 hr. Cells treated with RA and IAA were incubated for 16 hr at 0.25 µM and 500 µM, respectively.

## Cellular fractionation

Cell pellets (15 million) were lysed with Buffer A (0.1% Triton X-100, 10 mM HEPES pH 7.9, 10 mM KCl, 1.5 mM MgCl$_2$, 0.34 M sucrose, 10% glycerol, 1 mM DTT, 1× protease inhibitor mix) and incubated on ice for 8 min. Resulting nuclei were collected by centrifugation at 3000 × $g$ at 4°C. The supernatant was collected as the cytoplasmic fraction, and the nuclei were resuspended and washed in Buffer B (3 mM EDTA, 0.2 mM EGTA, 1 mM DTT, 1× protease inhibitor mix) twice. The nuclei were lysed with Buffer B+1% SDS on ice for 15 min before being mixed with 4× SDS (8% SDS, 200 mM Tris-HCl pH 6.8, 40% glycerol, 50 mM EDTA, 4% beta-mercaptoethanol, 0.04 % wt/vol bromophenol blue) in preparation for western blot.

## Generation of *Tbpl1* knock-out B8 cell line with CRISPR-Cas9

Two Guide RNAs (gRNAs) targeting the entire locus of Tbpl1 was designed using UC Santa Cruz CRISPR Guide RNA Design tool (http://crispor.tefor.net) and cloned into the pU6-(BbsI)_CBh-Cas9-T2A-mCherry (Addgene #64324). mESCs were grown to 50% confluency and then transfected with 1 µg of each Cas9 gRNA plasmid using Lipofectamine 2000 (Invitrogen 11668-019) according to the manufacturer's protocol. Cells were sorted 1 day after transfection gated for mCherry and GFP (H3.3 is endogenously tagged with GFP in all cell lines used), and 15,000 transfection positive cells were plated onto a 15 cm tissue culture plate. Cells were grown until individual colonies were visible and single colonies were transferred to a 96-well for screening. Seventy percent of each cell colony was used for genotyping, while the other 30% was grown for maintenance. For genotyping, cells were lysed using DirectPCR lysis reagent (Viagen Biotech #302C) according to the manufacturer's protocol and lysates were used in a screening PCR to identify edited cells. Identified clones were further validated using Sanger sequencing (not shown), qRT-PCR, and western blotting. A list of gRNAs and primers used for knock-out and screening are in *Supplementary file 6*.

## Generation of the overexpressed TRF2-HA cell line

mESCs were grown to 50% confluency and transfected using 1 µg of the TRF2-HA construct together with 1 µg of the Super Piggybac transposase plasmid (Gift from Tjian Lab) using Lipofectamine 2000 (Invitrogen 11668-019) according to the manufacturer's protocol. After 24 hr, 500 µg/mL of G418 (Fisher BP673-5) was added to select for successful random integration of the construct. Cell media containing antibiotics was refreshed daily until all negative control cells were dead.

## Antibodies for western blot

Primary antibodies: α-TBP 1:3000 (Abcam ab51841), α-H3K27 1:7000 (Cell Signaling Technologies C36B11), α-HA 1:1000 (EpiCypher 13-2010), α-Tubulin 1:7000 (Abcam ab6046), α-RPC7 1:2000 (Santa Cruz Biotechnology sc-48365), α-TAF4 1:2000 (Santa Cruz Biotechnology sc-136093), α-H3.3 1:5000 (Abnova, H00003021-M01), α-TRF2 1:3000 (Gift from Dr. László Tora), and α-TFIIA-γ 1:3000 (Santa Cruz Biotechnology, sc-374483). Secondary antibodies: IRDye 800CW Goat anti-mouse (Licor 926-32210), IRDye 800CW Goat anti-rabbit (Licor 925-32211), IRDye 680RD Goat anti-mouse (Licor 926-68070), or IRDye 680RD Goat anti-rabbit (Licor 926-68701).

## ChIP-qPCR

Control or IAA-treated cells were crosslinked in 1% formaldehyde for 5 min at room temperature and quenched with the addition of glycine to 0.125 M for 5 min at room temperature. Cells were harvested and washed twice in cold 1× PBS, centrifuged for 5 min at 1000 × $g$ at 4°C, and lysed on ice for 5 min with lysis buffer (1% SDS, 10 mM EDTA, 50 mM Tris-HCl pH 8.0, 1× protease inhibitors, 0.2 mM PMSF, 1 mM benzamidine). To fragment DNA, whole cell lysates were sonicated for 10 min (30 s on/30 s off intervals, Diagenode Bioruptor). Lysates were then pre-cleared using 50 µL of magnetic Protein G Dynabeads and incubated on an end-over-end rotator for 2 hr at 4°C. Samples were then placed on a magnetic rack and aliquoted (30% vol/vol for input, 35% vol/vol for α-TBP IP, 35% vol/vol for α-IgG IP). IP buffer (150 mM NaCl, 20 mM Tris-HCl pH 8, 1% Triton X-100, 1× protease inhibitor, 0.2 mM PMSF, 1 mM benzamidine) was added to the lysates to obtain a total volume of 1 mL per immunoprecipitation. Lysates were then either incubated with 5 µg of α-TBP (Abcam ab51841) or anti-Mouse IgG (Abcam ab46540) overnight at 4°C, and bound to 50 µL of magnetic Protein G Dynabeads while rotating for 2 hr at 4°C. Beads were washed thrice with wash buffer A (0.1% SDS, 1% Triton X-100,

2 mM EDTA, 150 mM NaCl, 20 mM Tris-HCl pH 8), twice with wash buffer B (0.1% SDS, 1% Triton X-100, 2 mM EDTA, 500 mM NaCl, 20 mM Tris-HCl pH 8), and thrice with wash buffer C (2 mM EDTA, 20 mM Tris-HCl pH 8, 10% glycerol). Crosslinked DNA was eluted thrice from the beads with 150 µL of elution buffer (1% SDS, 100 mM NaHCO$_3$) for 15 min at 65°C, with vortexing every 3 min for a total of 450 µL of eluate, followed by RNAseA treatment for 1 hr at 37°C. Twenty µg proteinase K and NaCl (final conc: 0.3 M) were added to each eluate and input, and crosslinks were reversed overnight at 65°C. Following proteinase K digestion, DNA was purified by phenol-chloroform isolation and resuspended in 50 µL of TE buffer. Two µL of ChIP DNA was used in each qPCR with Luna Universal qPCR Master Mix (M3003) according to the manufacturer's instructions. Samples were run on the Quant-Studio 3 Real-Time PCR System. Primers used in qPCR are listed in *Supplementary file 7*.

## NET-seq

NET-seq was performed as previously described (*Mayer and Churchman, 2016*) with the following modifications. Approximately 10–15 million mESCs were used in each fractionation. *Drosophila* S2 cells were added to each fractionation (3% by cell count) as spike-in for downstream analyses. Cells were lysed with cytoplasmic lysis buffer on ice for 7 min and nuclei were lysed in nuclei lysis buffer on ice for 2 min. Barcoded DNA linkers were riboadenylated as previously described (*Song et al., 2015*), with the following modifications. Adenylation reactions were performed in a reaction mixture (10 µL) containing 0.8 µg of barcoded DNA linker, 1× T4 RNA ligase buffer (NEB, B0216L), 2 mM ATP, 35% PEG, and 300 U of T4 RNA Ligase 1 (NEB, M0204L), incubated at 37°C for 8 hr, followed by 15 min of heat inactivation at 65°C. Riboadenylated barcoded DNA linkers were stored at –80°C until use. NET-seq library preparation was performed as described previously (*Mayer and Churchman, 2016*). cDNA containing sequences of a subset of sequenced snRNAs, snoRNAs, and rRNAs (*Supplementary file 5*) were specifically depleted using biotinylated DNA oligos (Integrated DNA Technologies). Sequencing was performed at the UBC Biomedical Research Centre with 55 bp single-end reads.

## Processing and alignment of NET-seq reads

NET-seq data was processed as described in *Mayer et al., 2015*. Reads were trimmed and aligned using STAR (v.2.7.3a). Reverse transcription mispriming events, PCR duplication events, and splicing intermediates were removed using custom Python scripts provided by the Churchman group (https://github.com/churchmanlab, *Mayer et al., 2015*). The bam files of biological replicates were merged using SAMtools and normalized by subsampling each sample by the number of aligned *Drosophila* reads across all samples using SAMtools (*Li et al., 2009*). BedGraph coverage files for gene plots were generated from normalized bam files with a custom Python script provided by the Churchman group (https://github.com/churchmanlab) and visualized by IGV. Bigwig coverage files for heatmaps and TSS plots were generated using deepTools with the following parameters: -bs 1 --Offset 1 (*Ramírez et al., 2016*). Heatmaps and TSS plots were generated using deepTools with the following parameters: -a 1000 -b 1000 -bs 10 followed by filterValues --max 50. Read counts across tRNAs were generated from normalized bam files using bedtools and analyzed with GraphPad Prism (*Quinlan and Hall, 2010*). Read counts used in edgeR analysis were generated from bam files using featureCounts (*Liao et al., 2014*). Reads were then imported into R Studio and analyzed using the DGE tool of the edgeR Bioconductor package (*Robinson et al., 2010*). First, DGEList was used to specify reads and gene names, followed by filtering (filterByExpr) and normalization by TMM (default). Once reads are normalized, the design matrix was built and dispersion was estimated to determine biological variation. The command glmFit was used to determine differentially expressed peaks/genes (DEG) and the command topTags was used to show the top genes. Scatter plots were generated using the plotMD function and significant genes were obtained along with the raw values for downstream analyses such as heatmap and table generation.

## CUT&Tag

CUT&Tag was performed as previously described (*Kaya-Okur et al., 2019*), but with the following modifications. Cells were harvested at room temperature and 100,000 mESCs were used per sample. With the EpiCypher pAG-Tn5 (EpiCypher EP151117), cryopreserved *Drosophila* S2 cells were spiked in at 20% (20,000 S2 per 100,000 mESCs) and 3% for the Pol II in the C64 6 hr IAA sample (*Figure 1—figure supplement 2C*) to account for a spike-in control, during the cell pelleting step

before washing. pAG-Tn5 from the Henikoff lab did not require a spike-in since it contained trace amounts of *E. coli*. Antibodies used include TBP (Abcam ab51841), RNA Pol II (Cell Signaling Technology D1G3K), RNA Pol III (Santa Cruz Biotechnology sc-21754), Rabbit anti-Mouse IgG (Abcam ab46540), α-H3K27me3 (Cell Signaling Technology C36B11), α-HA (EpiCypher 13-2010), α-TAF1 (Cell Signaling Technology D6J8B), α-TAF4 (Santa Cruz Biotechnology sc-136093), and α-TFIIA (Santa Cruz Biotechnology sc-25365). Secondary antibodies used include Guinea Pig anti-Rabbit IgG (Antibodies-Online ABIN101961) and Rabbit anti-Mouse IgG (Abcam ab46540). Secondary antibody incubation times were 45 min at room temperature. The Henikoff lab pA-Tn5 adapter complex or the commercial Epicypher pA-Tn5 was added at a final concentration of 1:250 or 1:20, respectively. Sequencing was performed at the UBC Biomedical Research Centre using the NextSeq500 75 cycles and the NextSeq2000 50 cycles.

## CUT&Tag analysis

Reads were mapped on mm10 genome build using Bowtie2 with the following parameters: `--no-unal --local --very-sensitive-local --no-discordant --no-mixed --contain --overlap --dovetail --phred33 -I 10 -X 9999`. PCR duplicate reads were kept as these sites may represent real sites from adapter insertion from Tn5 as per recommendation from the Henikoff lab. A normalization factor was determined from *E. coli* (normalized to the control) or *Drosophila melanogaster* (normalized to the control) alignment from Bowtie2 mapping and used to scale IAA-treated samples to control during generation of bigwig files. For the ChIPseqSpikeInFree pipeline, Pol II fastq files were subsampled to the lowest reads for each CUT&Tag run: 8 million reads (replicate 1&2) for C64 samples; 2 million reads (replicate 1&2); and 4 million reads (replicate 3&4) for B8 samples. Downstream analyses, heatmaps, TSS plots, Gene plots, k-means clustering were performed using IGV, DeepTools, and BedTools suite (*Ramírez et al., 2016*; *Quinlan and Hall, 2010*). ComputeMatrix from deeptools was done using binsize 10. Replicates were merged using the BigWigMerge tool, which sums the reads after normalizing treatment samples to the control samples of each run using *Drosophila* or *E. coli* spike-in. ChIPseqSpikeInFree provided its own normalization factors when running the pipeline. Alternate reference alignment was obtained from the UCSC browser tracks.

## DGE analysis using edgeR bioconductor

Scatter plots of differential gene expression (DGE) analysis were performed using the DGE tool from the edgeR bioconductor package. Starting subsampled bam files were used for all Pol II samples. Read counts were obtained using featureCounts command with paired end specificity for the entire gene and pseudogenes were removed (*Liao et al., 2014*). Reads were then imported into R Studio and analyzed using the DGE tool following the Bioconductor manual using default parameters (i.e. 5% FDR). First, DGEList was used to specify reads and gene names, followed by gene annotations (optional) then filtering and normalization by TMM (default option). Read counts below 20 were filtered using filterByExpr (min.count=20). Once reads were normalized, the design matrix was built and dispersion was estimated to determine biological variation. The command glmFit was used to determine DEG and the command topTags was used to show the top genes. Scatter plots were generated using the plotMD function and significant genes were obtained along with the raw values for downstream analyses such as heatmap and table generation.

## Gene ontology

GO was done on gene sets identified to be significant from the DGE analysis. The top HS and RA upregulated genes were input into http://geneontology.org/, filtered by molecular function for the *M. musculus* genome to obtain GO terms.

## Pearson correlation analysis

Pearson correlation analyses were performed using the multiBigwigSummary command from the DeepTools suite with the *bins* option and the *bedfile* option for the regions surrounding the TSS for TAF1, TAF4, and TFIIA (-/+1000 bp of the TSS). The Pearson correlation coefficients were plotted as a heatmap using the plotCorrelation command with the remove outliers and skip zeros options.

## Scatter plot read counts analyses and filtering

Read counts for scatter plots were obtained from bam files using the bedtools multicov command and plotted to the promoters of genes (–250 bp to TSS) (*Quinlan and Hall, 2010*). Read counts were then

normalized to the scaling factor, summed and plotted as is or log transformed before being plotted. Regression analysis was then performed to obtain the slope and $R^2$ value. Control vs. IAA samples were filtered by taking out genes that did not have any signal in the control (0 reads in the control samples). Fold changes between Pol II and TAF1/TAF4/TFIIA were filtered by genes with moderate to high levels of Pol II in the control (Pol II control summed reads 50 and below were excluded). Fold changes between TFIIA and TAF4 were filtered by moderate to high levels of TFIIA and TAF4 in the control (TFIIA and TAF4 control summed reads 50 and below were excluded).

## qRT-PCR

Cells were cultured until ~90% confluency on tissue culture-treated plates at 37°C in a 5% $CO_2$. After indicated treatments, cells were washed with 1× PBS, trypsinized and pelleted by centrifuging at 600 × $g$. RNA was then extracted from the pellet via Trizol extraction and concentrations were measured via DeNovix Nanodrop. Spike-in RNA was then added at 20% the sample amount. One µg of samples were then DNase treated following the Promega DNase kit (M6101). The entire sample was then reverse transcribed using the New Englands BioLabs LunaScript RT SuperMix kit (E3010). Samples were then diluted to 10 ng/µl and 20 ng was used for qRT-PCR experiment using the New Englands BioLabs Luna Universal qPCR Master Mix kit (M3003). Samples were then processed on the Quant-Studio 3 Real-Time PCR System.

## Immunofluorescence

Cells were grown on coverslips (Azn Scientific #ES0117650) that were pre-washed in 70% ethanol and coated with gelatin in tissue culture-treated six-well plates. After indicated treatments, cells were washed with 1 mL of PBS and fixed in 4% paraformaldehyde (UBC Chemical Stores #OR683105) for 15 min. After fixation, cells were washed and permeabilized with 1 mL 0.025% Triton-X for 5 min, followed by two 10 min washes with PBS. Samples were blocked by nutating in PBS 5% BSA for 30 min. TBP (Abcam ab51841) was diluted at 1:100 in PBS, and added to cells in coverslips for 1 hr. Samples are washed twice with PBS (5 min for each wash). Coverslips were then incubated in Alexa Fluor 594 (A11005) 1:100 for 1 hr, and washed with PBS twice (5 min for each wash). Samples are then incubated with DAPI (300 nM in PBS) for 5 min and washed twice with PBS (5 min for each wash). Coverslips were assembled using Vectashield mounting medium (BioLynx #VECTH1000). Fluorescent images were collected using the Leica DMI6000B inverted fluorescence microscope. Quantification was done through Fiji (*Schindelin et al., 2012*).

## CoIP

Cells were grown to 90% confluency on a 15 cm gelatinized plate. After indicated treatments, cells were washed with 1× PBS, trypsinized and pelleted by centrifuging at 600 × $g$. Cell pellets were then lysed in 1 mL of Lysis Buffer (200 mM NaCl, 25 mM HEPES, 1 mM $MgCl_2$, 0.2 mM EDTA, 0.5% NP-40, 1× Roche complete inhibitor, 0.2 mM PMSF, 1 mM benzamidine). The whole cell lysates were passed through a 25 G needle five times and incubated on an end-over-end rotator for 30 min at 4°C. Samples were spun at max speed at 4°C for 5 min. Supernatant was transferred to a new tube and the pellet was then digested as much as possible using 2 U of MNase in 1× MNase digestion buffer by nutating at 37°C for 30 min. Digested pellet was centrifuged at max speed and the new supernatant was added to the previous supernatant. Lysates were then pre-cleared using 20 µL/sample of Protein G Dynabeads and incubated on an end-over-end rotator for 2 hr at 4°C. Samples were then placed on a magnetic rack and aliquoted (5% vol/vol for input, 45% vol/vol for α-TAF4 IP, 45% vol/vol for α-IgG IP). Pull-down was performed on the cell lysates by adding 3 µg of α-TAF4 (Santa Cruz Biotechnology sc-136093) or mouse α-IgG (Santa Cruz Biotechnology sc-2025) crosslinked to Protein G Dynabeads and incubating on an end-over-end rotator overnight at 4°C. The next day, beads were washed six times with 600 µL Lysis Buffer. Proteins were eluted from the beads with 50 µL of 0.1 M glycine (pH 2.5) for 30 min and neutralized with 50 µL 1 M Tris-HCl (pH 8.0). Elution was repeated two more times and all eluates were combined. Five percent vol/vol of the final eluate was taken to assess the IPs by western blot, while the rest of the co-IP eluate was flash frozen with $LN_2$ for MS analysis.

## Mass spectrometry

Eluates were purified and digested following the Single-pot, solid-phase-enhanced sample preparation (SP3) method (https://doi.org/10.1038/s41596-018-0082-x) as follows. Eluates were reduced using 2.55 µL of 1 M dithiothreitol (DTT, 10 mM final concentration) for 30 min at 37°C, followed by alkylation using 25.5 µL of 500 mM chloroacetamide (CAA, 50 mM final concentration) for 30 min at room temperature in the dark, and quenched with 14.2 µL of 1 M DTT (50 mM final concentration) for 5 min at room temperature. Sera-Mag Speed Beads were prepared by mixing in equal volume (10 µL hydrophilic and 10 µL hydrophobic per sample), rinsed with water twice, and resuspended in half the volume (total of 10 µL mixed beads per sample) in water. Prepared beads were added to the sample and protein binding was facilitated by adding 100% ethanol (EtOH) to a final concentration of 80%. Samples were incubated for 5 min in a thermomixer at room temperature. Beads were magnetically isolated from the supernatant and proceeded to wash three times with 1 mL 90% EtOH. Excess EtOH presented in beads was removed by brief centrifugation and aeration. Beads were then resuspended in 100 mM ammonium bicarbonate and digested with trypsin/LysC at 1:75 protease:protein (w/w) ratio in 37°C thermomixer overnight.

Peptide digests were magnetically isolated from beads and acidified to pH 3–4 with trifluoroacetic acid (TFA). Peptides were desalted using self-packed C18 StageTips with the following steps: tips were conditioned with methanol, elution buffer and equilibrate with wash buffer before loading acidified peptide samples (loaded for a total of two times), peptides bound on disks were washed with washing buffer twice before eluting in 60 µL of elution buffer. Washing buffer consisted of 0.1% TFA in water and the elution buffer consisted of 60% acetonitrile in 0.1% formic acid (FA). All the above steps were performed in 100 µL volume and spun at 1200 × $g$ unless stated otherwise. Eluted peptides were dried in a Speed Vac.

Mass spectrometry analysis was performed on a Q Exactive HF Orbitrap mass spectrometer coupled to an Easy nLC-1200 Liquid Chromatography system (Thermo Scientific). Dried peptides were resolubilized in 0.1% FA in water spiked with iRT Standard (Biognosys) and peptide concentration was measured by nanodrop. One µg of each sample was loaded onto a 50 cm µPAC HPLC Column (Thermo Scientific) maintained at 45°C and separated by a 1 hr gradient with Buffer B ramping from 4% to 27%, at 300 nL/min gradient flow. Buffer A was 2% ACN in 0.1% FA in water and Buffer B was 95% ACN in 0.1% FA in water. MS acquisition was performed with a full-scan MS spectrum between 300 and 1650 m/z, resolution of 120,000, AGC target of 3e6, and maximum IT of 60 ms followed by DIA MS2 acquisition using 24-variable isolation windows at 30,000 resolution, AGC target of 3e6, auto maximum IT, and stepped NCE of 25.5, 27, 30.

Acquired DIA data was analyzed using directDIA analysis in Spectronaut (version 14.10.201222.47784). The Spectronaut Pulsar search settings were as follows: Enzyme and digest type was Specific Trypsin/P with two missed cleavages. Carbamidomethyl (C) was set as fixed modification; acetyl (N-term) and oxidation (M) were defined as variable modifications. Spectra were matched against the curated mouse proteome fasta from Uniprot (obtained 2021-03-01) with 17,063 entries. False discovery rate (FDR) of 0.01 was set for peptide spectra matching, peptide and protein group identification and the precursor PEP cutoff was set at 1E-10. Missing peptides identifications were imputed using the 'Global Imputing' strategy with a Qvalue percentile cutoff of 0.2. The 'minor group quantity' was calculated as the mean precursor quantity and the 'major group quantity' was calculated as the median of 1–6 peptide quantities. Data was normalized across runs using a Qvalue percentile cutoff of 0.8. All data has been deposited to ProteomeExchange (PXD034171) through MassIVE (MSV000089562).

## Gel filtration

Nuclei was isolated from ~25 million cells per sample as previously described (**Mayer and Churchman, 2016**) with the following modifications. Nuclei was resuspended in glycerol-free nuclei resuspension buffer (20 mM Tris HCl pH 8, 75 mM NaCl, 1× protease inhibitors, 0.1 mM PMSF) and lysed with equal volume of urea-free nuclei lysis buffer (1% NP-40, 50 mM HEPES, 325 mM NaCl, 1× protease inhibitors, 0.1 mM PMSF, 2 mM MgCl$_2$, 50 µg/mL benzonase) on an end-over-end rotator at 4°C for 30 min. Samples were spun at 21,000 × $g$ at 4°C for 5 min and supernatant was transferred to a new tube for gel filtration analysis.

A Superose 6 Increase 10/300 GL column was pre-equilibrated with running buffer (10 mM Tris HCl pH 8, 0.5% NP-40, 25 mM HEPES, 200 mM NaCl, 0.1 mM PMSF) prior to use; 0.5 mg of nuclear lysates

were loaded onto the Superose 6 column using the AKTA Purifier FPLC system (GE Healthcare) and run at 0.5 mL/min. Filtered proteins were detected by absorbance at 280 nm, and 47×500 µL fractions were collected for subsequent analysis by western blot.

## Acknowledgements

We thank Dr. Steven Henikoff for generously providing the pA-Tn5 enzyme, and Dr. László Tora for generously providing the anti-TRF2 antibody. We thank Reid Warsaba, Jibin Sadasivan, and Dr. Eric Jan for providing S2 cells and for providing insight on qRT-PCR S2 spike-in normalization. We thank R Vander Werff and T Stach (BRC-seq, UBC) for Illumina sequencing and S Flibotte (LSI Bioinformatics facility, UBC) for implementation of NET-seq analyses. This work was supported by Life Sciences Institute Cores (LSI Imaging, ubcFLOW, and QPCR Core), supported by the UBC GREx Biological Resilience Initiative. For insightful comments on the manuscript, we thank Drs. Annie Ciernia and Ethan Greenblatt. MAB is a postdoctoral fellow supported by the Sigrid Jusélius Foundation. ACU is supported by the BC Children's Hospital Foundation. PFL is a Canada Research Chair and Michael Smith Foundation for Health Research Scholar. SST is a Canada Research Chair and Michael Smith Foundation for Health Research Scholar. Funding: This work was supported by the Canadian Institutes for Health Research Project Grant award to SST (PJT-162289). The National Sciences and Engineering Research Council Discovery Grant award to SST (RGPIN-2020-06106).

## Additional information

### Funding

| Funder | Grant reference number | Author |
| --- | --- | --- |
| Canadian Institutes of Health Research | PJT-162289 | Sheila S Teves |
| Natural Sciences and Engineering Research Council of Canada | RGPIN-2020-06106 | Sheila S Teves |

The funders had no role in study design, data collection and interpretation, or the decision to submit the work for publication.

### Author contributions

James ZJ Kwan, Conceptualization, Funding acquisition, Investigation, Visualization, Methodology, Writing - original draft, Writing - review and editing; Thomas F Nguyen, Conceptualization, Investigation, Visualization, Methodology, Writing - original draft, Writing - review and editing; Anuli C Uzozie, Investigation, Visualization, Methodology; Marek A Budzynski, Funding acquisition, Investigation, Visualization, Methodology, Writing - review and editing; Jieying Cui, Investigation, Visualization, Writing - review and editing; Joseph MC Lee, Filip Van Petegem, Methodology; Philipp F Lange, Funding acquisition, Investigation, Visualization, Writing - review and editing; Sheila S Teves, Conceptualization, Supervision, Funding acquisition, Visualization, Writing - original draft, Project administration, Writing - review and editing

### Author ORCIDs

James ZJ Kwan http://orcid.org/0000-0003-4467-1749
Thomas F Nguyen http://orcid.org/0000-0003-3269-0846
Philipp F Lange http://orcid.org/0000-0003-1171-5864
Sheila S Teves http://orcid.org/0000-0002-1220-2414

### Decision letter and Author response

Decision letter https://doi.org/10.7554/eLife.83810.sa1
Author response https://doi.org/10.7554/eLife.83810.sa2

# Additional files

## Supplementary files

- Supplementary file 1. Top upregulated/downregulated genes extracted from differential gene expression (DGE) analysis of polymerase II (Pol II) cleavage under targets and tagmentation (CUT&Tag) for all genes in control and IAA-treated C64 mESCs based on *Figure 1F*.

- Supplementary file 2. Raw values of a few top upregulated HS genes extracted from differential gene expression (DGE) analysis of polymerase II (Pol II) cleavage under targets and tagmentation (CUT&Tag) on all genes in control vs. HS-treated C64 mESCs.

- Supplementary file 3. Gene ontology analysis of the top upregulated genes extracted from differential gene expression (DGE) analysis of polymerase II (Pol II) cleavage under targets and tagmentation (CUT&Tag) in control vs. HS-treated C64 mESCs from *Figure 1—figure supplement 3A*.

- Supplementary file 4. Raw values of a few top upregulated RA genes extracted from edgeR differential gene expression (DGE) analysis of polymerase II (Pol II) cleavage under targets and tagmentation (CUT&Tag) on all genes in control vs. RA-treated C64 mESCs.

- Supplementary file 5. Biotinylated snRNA, snoRNA, lncRNA, and rRNA depletion oligonucleotides for NET-seq library construction. Oligonucleotides were modified with the addition of a 5' biotin group and purified by HPLC.

- Supplementary file 6. Primers used to knock-out *Tbpl1*, confirm *Tbpl1* knock-out, and to test *Gapdh* mRNA levels via qRT-PCR.

- Supplementary file 7. Primers used to test TATA-box binding protein (TBP) depletion via ChIP-qPCR.

- MDAR checklist

## Data availability

All sequencing data have been deposited in Gene Expression Omnibus (Accession number GSE172401). All mass spectrometry data have been deposited to ProteomeExchange (PXD034171) through MassIVE (MSV000089562). All other data are available in the manuscript or in the supplementary materials.

The following datasets were generated:

| Author(s) | Year | Dataset title | Dataset URL | Database and Identifier |
|---|---|---|---|---|
| Kwan JZJ | 2021 | RNA Polymerase II transcription independent of TBP | https://www.ncbi.nlm.nih.gov/geo/query/acc.cgi?acc=GSE172401 | NCBI Gene Expression Omnibus, GSE172401 |
| Kwan JZJ | 2021 | RNA Polymerase II transcription independent of TBP | https://massive.ucsd.edu/ProteoSAFe/private-dataset.jsp?task=c23b3528108340d58df6b4544eb39e4c | Massive, MSV |

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
