## [Editor Report]

This important study employs auxin-induced degradation to provide compelling evidence that the TATA-binding protein (TBP) is not required for ongoing RNA polymerase II transcription nor heat-shock or retinoic acid-induced transcription in murine embryonic stem cells, but is essential for RNA polymerase III transcription. Furthermore, the study shows that TBP-independent TFIID complexes are assembled and present at the transcription start sites of polymerase II-transcribed promoters. This work will be of broad general interest to the regulation of gene expression field.

---

## [Decision Letter]

**Decision letter after peer review:**

Thank you for submitting your article "RNA Polymerase II transcription independent of TBP" for consideration by *eLife*. Your article has been reviewed by 3 peer reviewers, including Irwin Davidson as the Reviewing Editor and Reviewer #1, and the evaluation has been overseen by Kevin Struhl as the Senior Editor.

Your paper has been discussed amongst the referees, the reviewing editor Irwin Davidson and senior editor Kevin Struhl. The general opinion was that this study is of interest to *eLife*, but requires a thorough revision before publication can be considered. The referee's comments are included below. Several issues require particular attention.

1. The single molecule dynamics experiments are a very indirect way to assess TAF/TFIID association with promoters and since the experiments were performed with ectopically expressed TAFs, the interpretation requires a considerable number of additional controls to show that the proteins are integrated quantitatively into TFIID and not interacting with chromatin individually or in partial TFIID subcomplexes. We suggest that Figure 6 be removed altogether and that the data in Supplementary Figure 9 be moved into the main text. However, there are concerns with the reproducibility of the TAF1 experiments and it is contradictory to claim that a single TBP-free TFIID binds to promoters and at the same time present data showing that TAF1 binding is comparable to the control, whereas that of TAF4 is strongly reduced. All questions in the referee's comments on quantification of replicates require particular attention.

2. Given the important implications of the study, we request that the Cut&Tag data be consolidated by a TBP ChIP-seq with and without auxin treatment to fully ensure that TBP has been depleted from promoters. Similarly, it is essential to assess promoter recruitment of TFIIA and TFIIB upon TBP depletion.

3. In addition, assessment of TFIID assembly in absence of TBP by gel filtration would also strengthen the study.

4. Lastly, the effect of TBP depletion on Pol I transcription should be addressed and the text revised to tone down the interpretation as it is virtually impossible to exclude the idea that residual TBP is sufficient to ensure PIC formation.

We look forward to a revised version where these issues and those in the referee's reports have been addressed.

*Reviewer #1 (Recommendations for the authors):*

1. Concerning the TBP-independent TFIID complex. The authors show that in absence of TBP other TFIID subunits can coprecipitated with TAF4. The authors should perform gel filtration experiments to determine whether the residual complex is a single TFIID without TBP or whether it has dissociated into diverse TAF4-containing (or without TAF4 that are not taken into account by the MS experiments) subcomplexes of lower molecular mass. None of the experiments in the manuscript address the nature of the TFIID that is recruited to promoters in absence of TBP.

2. A second major issue is the TBP-independent recruitment of TFIIA and TFIIB to promoters. It is believed that both these proteins interact with TBP and that TBP-induced DNA bending aides in their promoter recruitment. How is their occupancy affected in the absence of TBP? Also, TFIIA shows a strong interaction with TRF2, what happens to TFIIA recruitment in the TBP-depleted TRF2 KO cells?

3. The authors argue that TFIID is recruited to promoters in absence of TBP and they show data for TAF1 and TAF4. Can the authors test whether Pol II transcription in general or at specific promoters is more sensitive to loss (siRNA knockdown) of certain TAFs in absence of TBP that in its presence.

*Reviewer #2 (Recommendations for the authors):*

(lines 30-32) The conformational changes induced by TBP within TFIID trigger

the binding of TFIIA and TFIIB (3), followed by TFIIF and Pol II, which further stabilizes the partial PIC (3). These statements are incorrect and the citation inappropriate.

This is reference (3):Tolić-Nørrelykke, S. F., Rasmussen, M. B., Pavone, F. S., Berg-Sørensen, K., and Oddershede, L. B. (2006) Stepwise Bending of DNA by a Single TATA-Box Binding Protein.830, Biophysical Journal. 90, 3694-3703

The authors may be referring to Roeder, R. G. 1996: The role of general initiation factors in transcription by RNA polymerase II. Trends Biochem. Sci. 21:327-335., which is what the (3) authors are mentioning in their introduction. But even in that case, the first statement is incorrect!

Also reference (4) does not seem at all the most appropriate. Roeder 1996 would be better.

Reference (5) is also totally inappropriate!!

In general, I would ask the authors to double check their citations!!

Concerning TFTC , also known as STAGA, now they are considered to both correspond to the SAGA complex.

(line 267 and beyond) The visualization of overexpressed halo-tagged TAFs does not seem the most appropriate way to look at the dynamics of the TAF-containing complexes. The proper experiment would have been to tag the chromosomal copies.

(lines 277-81) By bound TAF7 molecules, do they mean bound to TFIID or bound to DNA, or both or a mixture? Same for TAF4. If they refer to DNA bound, do they know if they are bound to DNA as part of TFIID or by themselves? With the overexpression, this is a little more difficult to deconvolute and to interpret cleanly.

For the residency time on DNA, how can the authors distinguish if the complex is bound to core promoter DNA or tethered to the +1 nucleosome via reader domains? And, again, is this occurring as individual subunits or in the context of the complex or a mixture? While the two different values for the two subunits make it less likely that both of them are binding as part of a full TFIID, the different possibilities should be kept in mind when interpreting the results.

On that note, TAF4 has activator binding domains, so part of its dynamics can be affected by that, and this has to also be seen in the context of the overexpression

That fewer TAF7 binds to chromatin when TBP is depleted, but those bound stay longer, again needs to be put in the context of the overexpression and of the partial depletion of TBP

If the authors are assuming that TAF7 and TAF4 somehow act differently because they are present in different TFIID complexes, should the immuno-purification via TAF4 show substochiometric amounts of some of the other components? Can they check that?

Can the authors add a statement about how the CUT&Tag experiments and the single particle experiments relate to each other quantitatively?

(316) "based on the crystal structure of the TFIID complex" – should be "cryo-EM structure". On that note, there are two copies of TAF4, and based on structural studies, each of them can be close or far from TBP depending on whether TBP has or not bound the core promoter. By the way, TAF1 is even closer (directly interacting with TBP through its N-terminus) in free TFIID, but further that at least one copy of TAF4, once TBP engages the core promoter.

(327-8) "In contrast to TAF4, the largest subunit maintains its binding to

promoters in the absence of TBP depletion." This does not make sense. Do they mean "in the case of TBP depletion"?

Finally, I am not an expert on the genomic analyses included, so I cannot judge the quality of the data or its interpretation beyond what the authors propose. So, I look forward to discussing the paper with the rest of the reviewers.

*Reviewer #3 (Recommendations for the authors):*

– The authors should show and quantify the depletion of TBP in all cellular compartments, cytoplasm, nuclear soluble and the chromatin associated fractions after 6 and 16 hours of auxin treatment. In the Result section and in the legend of Figure 1A the authors should indicate which type of extract they are analysing on the western blot.

– As the authors state that after the used auxin treatment about 10-20% TBP is remaining, the numerous statements giving the impression that TBP has no role in Pol II transcription should be tuned down throughout the whole manuscript. What if the remaining 10000-20000 TBP molecules can still work in Pol II transcription because they are very dynamic?

– How many TBP molecules work in Pol I, Pol II and Pol III GTF complexes? If there are many more TFIIIB complexes (containing TBP) than TFIID or SL1 complexes, it is logical that the first effect one will see after TBP depletion is the decrease of Pol III transcription. Can the authors quantify the number of TBP-containing TFIIIB complexes versus TFIID and SL1 complexes?

– Along the same lines, the authors should also test the effect of TBP depletion on Pol I transcription.

– How dynamic is TBP in comparison to Pol I, Pol II and Pol III TAFs and/or Brf1? What are the corresponding DNA residency times of these factors?

– From Figure S2A it is evident that the most highly depleted genes for TBP binding are the most highly transcribed (high Pol II and NET-seq signals). Could TBP be more dynamic there? The same correlation is visible with TAFs (Figure 7D-F). Could TAFs favor TBP dynamics at promoters (making it more "depletable", maybe through TAF1 TAND)? Authors should show, side-by-side, the CUT&Tag occupancy heatmaps for TBP, TAF1, TAF4, Pol II and NET-seq with genes ordered by TBP occupancy (+/- IAA).

– Figure Supp. 5G (not cited in the text) shows that under TBP and TBPL1 (TRF2) co-depletion Pol II occupancy increases about 5 fold (in replicate 2). How do the authors explain this increase? Could this be a normalization artefact? How is it possible that authors conclude from these conflicting replicate experiments that Pol II occupancy is not affected globally (lines 242-243)?

– Experiments described in Figure 6 are lacking several key controls before they can be interpreted. First the authors would need to demonstrate that ALL the "overexpressed" Halo-tagged TAF4 or Halo-TAF7 incorporate into endogenous TFIID complexes. Otherwise the presented Spot-On measurements are meaningless, because the authors may simply measure the behavior of free TAFs. What is the sense of deriving/showing the % of free/complexed/bound states in an overexpressed system (Figure 6)? No comparison can be made among different subunits (TAF7 vs TAF4) as attempted by the authors in the discussion (line 403, "our SPT results indeed show that TAF7 is much less DNA-bound..")

– To normalize their CUT&Tag data, the authors used a spike-free normalization method. This method has only been validated for very abundant histone mark ChIPs, not for Pol II. It is not clear why the authors think that spike-free normalization method is needed for Pol II CUT&Tag normalization. In Supp. Fig1F, the authors show two Pol II CUT&Tag replicates with *E. coli* spike-in and the replicates are quite different. Why do the authors think that the normalized data showing Pol II decrease in IAA condition is not correct?

– How do authors interpret the widespread CUT&Tag signals they obtain on gene bodies for TAFs (especially TAF1) and partially also for TBP. Do these correspond to non-specific, transient binding events? If yes one would expect that a portion of the signal localized around promoter regions would also be of the same nature.

– Heat-shock & RA induction might not be the best systems to test for the "reinitiation" hypothesis since TBP might already be promoter bound prior to activation. Indeed, there is a TBP peak even in non-treated condition in the tracks shown in Figure 3 and S3.

– The RA treatment experiment has to be repeated similarly to the HS experiment, first 6h auxin treatment to degrade TBP and then RA treatment. At the beginning of the RA treatment TBP is still present in the cells making the interpretation of these experiments impossible.

– In several endogenous TAF IPs TBP has been found to be sub-stoichiometric, could this mean that TBP is not a stable component of mammalian TFIID complexes? TBP dynamics are also regulated by NC2 and BTAF1 complexes (PMID: 20627952), that could influence the interpretation of the obtained result.

– The discussion sentence "The IP-MS data showing that the TFIID complex forms upon TBP depletion is surprising" is misleading, since at structural level TBP depletion is not expected to deeply affect any TFIID subunit or complex integrity.

– "The altered dynamics of TAF4 upon TBP depletion may be due to its close proximity to TBP, based on the crystal structure of the TFIID complex". There is no crystal structure of TFIID. TAF4 is instead involved in the interaction with TFIIA and, quite indirectly, with DNA-bound TBP.

---

## [Author Response]

Essential revisions:1. The single molecule dynamics experiments are a very indirect way to assess TAF/TFIID association with promoters and since the experiments were performed with ectopically expressed TAFs, the interpretation requires a considerable number of additional controls to show that the proteins are integrated quantitatively into TFIID and not interacting with chromatin individually or in partial TFIID subcomplexes. We suggest that Figure 6 be removed altogether and that the data in Supplementary Figure 9 be moved into the main text. However, there are concerns with the reproducibility of the TAF1 experiments and it is contradictory to claim that a single TBP-free TFIID binds to promoters and at the same time present data showing that TAF1 binding is comparable to the control, whereas that of TAF4 is strongly reduced. All questions in the referee's comments on quantification of replicates require particular attention.

We agree that the over-expression Halo-TAF SPT analyses may be too preliminary at this time. We have removed the original Figure 6 in the revised manuscript.

We also agree that the replicates for TAF1 CUT&Tag are not as concordant as other experiments. To address this, we have performed two more replicates of TAF1 CUT&Tag in control and TBP-depleted conditions, for a total of 4 replicates. Plots for each replicate are now shown in new Figure 6—figure supplement 1A-B. All four replicates are consistent with each other, confirming that TAF1 binding is largely unchanged when TBP is depleted.

Based on the TAF4 IP-MS and TAF1 and TAF4 CUT&Tag, we conclude that the complex can form even upon TBP depletion, and that some subunits can bind to promoters after TBP depletion. We wish to clarify that we do not conclude that “a single TBP-free TFIID binds to promoters”. Indeed, in the original manuscript, we speculated in the Discussion section that TFIID may not behave as a single monolithic complex. Rather, it could behave as a dynamic hub such that subunits would come and leave the ‘complex’ with different dynamics. This speculative model of a dynamic complex would be consistent with Halo-TAFs having different residence times as measured by SPT. Also, it would also be consistent with TAF1 and TAF4 having different CUT&Tag profiles after TBP depletion if they have different residence times on DNA.

In the revised version, we have removed references to the TAF dynamics since we have removed the SPT data. Instead, we include references to previous studies showing that TAFs form subcomplexes and may assemble into TFIID in a modular fashion. We also include new gel filtration data showing that TAF4 does not form other complexes when TBP is depleted, and new analysis of the TAF4 IP-MS data showing no change in the relative stoichiometries of the TAF subunits when TBP is depleted (see response to consensus comment 3). These new data and analyses are consistent with our conclusion that the full TFIID complex forms when TBP is depleted. However, biochemical detection of the full complex does not necessarily provide clear information on how the subunits bind to DNA. The behavior of individual subunits would be a combination of the dynamics of its association with the full complex, and the rate of dissociation from the DNA. Techniques such as CUT&Tag would be heavily influenced by how long TFs bind to DNA (residence time), which is inversely proportional to the rate of dissociation. Therefore, we argue that the IP-MS and CUT&Tag data do not provide contradictory information. Rather, they provide information on different aspects of TAF behaviors.

We also note that new studies using live cell imaging are beginning to show that biochemically isolated complexes can have very dynamic behaviors in live cells. These new studies are leading to a revised concept of a ‘complex’ in cellular context, including the CTCF/Cohesin complex in mESCs (Hansen et al. 2019 *eLife*; Gabriele et al. 2022 *Science*). With this new framework in mind, it is possible for a biochemical complex to form (TFIID as measured by IP-MS) while also having distinct subunit dynamics that manifests as different binding profiles measured by CUT&Tag.

2. Given the important implications of the study, we request that the Cut&Tag data be consolidated by a TBP ChIP-seq with and without auxin treatment to fully ensure that TBP has been depleted from promoters. Similarly, it is essential to assess promoter recruitment of TFIIA and TFIIB upon TBP depletion.

We agree, and we have performed ChIP-qPCR on select promoters to assess TBP occupancy after depletion. We observed decreases in ChIP levels down to IgG background levels, and this data is now included in the revised Figure 1—figure supplement 1D and is referenced in the main text. Importantly, the ChIP-qPCR results are consistent with the CUT&Tag data, which shows over 90% depletion at promoter regions genome-wide. Taken together, we have provided five orthogonal experiments to validate the TBP depletion system – Western, IF, ChIP-qPCR, CUT&Tag, and IP-MS.

We also agree that assessing TFIIA and TFIIB promoter occupancy after TBP depletion would be informative. Towards addressing this comment, we have performed spike-in normalized CUT&Tag using anti-TFIIA and TFIIB. The TFIIA CUT&Tag was successful, but the TFIIB antibodies did not produce signal above IgG background (we tried both Santa Cruz Biotechnology α-TFIIB (C-18) sc-225 and Cell Signaling Technology α-TFIIB 2F6A3H4). We aim to examine TFIIB through other means and other PIC members in the future. For now, we have included the results for TFIIA CUT&Tag in control and TBP-depleted conditions in revised Figure 6, with replicate information now shown in new Figure 6—figure supplement 2. A summary of the findings are now included in the main manuscript. Briefly, we found that TFIIA binding to promoters is decreased by ~50-60% relative to control conditions when TBP is depleted. We also performed a reanalysis of the TAF4 IP-MS data, which shows that the interaction between TFIIA and TAF4 is decreased when TBP is depleted (Figure 6—figure supplement 2A), even though we observed no change in TFIIA protein levels (Figure 6—figure supplement 2A). Consistent with the lack of effect on Pol II occupancy and activity, however, the genes that experience loss of TFIIA binding show no change in Pol II binding (Figure 6—figure supplement 3A,C). Therefore, we conclude that TBP depletion affects distinct members of the PIC, but that the system remains sufficiently robust to allow for Pol II transcription.

3. In addition, assessment of TFIID assembly in absence of TBP by gel filtration would also strengthen the study.

We agree that determining the nature of TFIID without TBP would be informative. As we discussed in our response to consensus comment #1, we have performed a gel filtration experiment on nuclear lysates in control and TBP depleted cells. We detected no changes in TAF4 elution patterns between control and TBP depleted conditions, and this data is now presented in revised Figure 5—figure supplement 1. We chose to analyze the nuclear fraction to specifically focus on the complex that could form on promoters, which means that we are blind to the existence of subcomplexes found in the cytoplasm. However, by analyzing the IP-MS data, we observe that the relative stoichiometries of TAF subunits do not change when TBP is depleted. Therefore, regardless of the nature of any subcomplexes forming in the cytoplasm, we show that their interactions do not change when TBP is depleted. We have now included this analysis in Figure 5—figure supplement 1.

4. Lastly, the effect of TBP depletion on Pol I transcription should be addressed and the text revised to tone down the interpretation as it is virtually impossible to exclude the idea that residual TBP is sufficient to ensure PIC formation.We look forward to a revised version where these issues and those in the referee's reports have been addressed.

To determine the effects of TBP depletion on Pol I transcription, we examined two orthogonal data sets: (1) NET-seq data; and (2) previously published chromatin-associated RNA-seq on control and TBP-depleted mESCs (PMID: 29939130). For both data sets, we quantified the signal for all rRNA genes in the mm10 genome in control and TBP-depleted conditions, and plotted the data as a scatter plot (see Author response image 1). In both data sets, the data for all rRNA genes fall mostly on the diagonal. These results suggest that, like Pol II, Pol I transcription may not be affected by acute TBP depletion. However, a more stringent examination is needed, much like we have done for Pol II, but this is beyond the scope of this manuscript and we look forward to pursuing this new direction in future studies.

**Author response image 1. sa2fig1:** 

We agree with the reviewers that the interpretation of our results need to be restricted. Throughout the manuscript, we have revised the text to adhere strictly to the conclusion that acute TBP depletion does not affect Pol II transcription. Furthermore, in the Discussion section, we have provided a thorough discussion on the inherent limitations of the mAID system and what it could mean for our study. We also added that although TBP depletion does not affect Pol II transcription, our system does not provide information on what TBP is doing when bound to promoters. Specifically, we state: “However, these results do not provide information on the functional role of TBP when recruited to Pol II promoters. One possibility is that Pol II transcription has evolved to have multiple redundancies such that removal of one factor (TBP) may not fully disrupt the whole system. Another possibility is that the transient nature of the embryonic cell state, with its unique chromatin landscape, enables Pol II transcription without TBP.” (Discussion section).

Reviewer #1 (Recommendations for the authors):1. Concerning the TBP-independent TFIID complex. The authors show that in absence of TBP other TFIID subunits can coprecipitated with TAF4. The authors should perform gel filtration experiments to determine whether the residual complex is a single TFIID without TBP or whether it has dissociated into diverse TAF4-containing (or without TAF4 that are not taken into account by the MS experiments) subcomplexes of lower molecular mass. None of the experiments in the manuscript address the nature of the TFIID that is recruited to promoters in absence of TBP.

Please see our response to consensus comment #1 and 3 above.

2. A second major issue is the TBP-independent recruitment of TFIIA and TFIIB to promoters. It is believed that both these proteins interact with TBP and that TBP-induced DNA bending aides in their promoter recruitment. How is their occupancy affected in the absence of TBP? Also, TFIIA shows a strong interaction with TRF2, what happens to TFIIA recruitment in the TBP-depleted TRF2 KO cells?

As we discussed above, we have performed TFIIA and TFIIB CUT&Tag in control and TBP-depleted mESCs. Unfortunately, the TFIIB antibody was not successful in the assay, but the TFIIA CUT&Tag shows strong signal at the TSS of all active genes (revised Figure 6) in control conditions. TBP-depletion led to a decrease in TFIIA binding by 50-60%, as measured by the genome-wide average profile. Consistent with the TAF4 results, the decrease in TFIIA binding when TBP is depleted suggests that: (1) Pol II is robust enough that a decrease in binding of some of the GTFs does not disrupt its activity; and (2) although Pol II transcription is unaffected, the PIC may undergo major changes in dynamics and/or DNA-binding kinetics. These findings are now summarized in the revised manuscript. Furthermore, a reanalysis of the TAF4 IP-MS data shows that the interaction between TAF4(TFIID) with TFIIA is significantly reduced when TBP is depleted. This data is now presented in the new Figure 6—figure supplement 2A.

Since TRF2 is not functionally redundant to TBP, we believe that what happens to TFIIA when TBP is depleted in the TRF2 KO cells is outside the scope of this study. However, we are excited to pursue other TRF2-specific mechanisms in future studies.

3. The authors argue that TFIID is recruited to promoters in absence of TBP and they show data for TAF1 and TAF4. Can the authors test whether Pol II transcription in general or at specific promoters is more sensitive to loss (siRNA knockdown) of certain TAFs in absence of TBP that in its presence.

We would love to test combinatorial knockdowns of TAFs with TBP. However, as there are 13 subunits, the scale is currently beyond what we can accomplish in a revision for this manuscript. However, we plan to systematically test all 13 subunits in future studies.

Reviewer #2 (Recommendations for the authors):(lines 30-32) The conformational changes induced by TBP within TFIID triggerthe binding of TFIIA and TFIIB (3), followed by TFIIF and Pol II, which further stabilizes the partial PIC (3). These statements are incorrect and the citation inappropriate.This is reference (3):Tolić-Nørrelykke, S. F., Rasmussen, M. B., Pavone, F. S., Berg-Sørensen, K., and Oddershede, L. B. (2006) Stepwise Bending of DNA by a Single TATA-Box Binding Protein.830, Biophysical Journal. 90, 3694-3703The authors may be referring to Roeder, R. G. 1996: The role of general initiation factors in transcription by RNA polymerase II. Trends Biochem. Sci. 21:327-335., which is what the (3) authors are mentioning in their introduction. But even in that case, the first statement is incorrect!Also reference (4) does not seem at all the most appropriate. Roeder 1996 would be better.Reference (5) is also totally inappropriate!!In general, I would ask the authors to double check their citations!!

We agree with the reviewer and apologize for the oversight. We have checked and revised the references throughout the manuscript.

Concerning TFTC , also known as STAGA, now they are considered to both correspond to the SAGA complex.

We agree and we have modified the text to reflect this change: “TFTC was subsequently shown to correspond to the SAGA complex…” (pg. 3).

(line 267 and beyond) The visualization of overexpressed halo-tagged TAFs does not seem the most appropriate way to look at the dynamics of the TAF-containing complexes. The proper experiment would have been to tag the chromosomal copies.

We agree that our over-expression system is limited in capacity, and that endogenous tagging is the more rigorous approach. Since our imaging experiments are somewhat preliminary, we have removed these experiments from the manuscript. Because of this, comments 2-8 are no longer applicable. We hope to use these comments to better design imaging experiments for future studies.

(lines 277-81) By bound TAF7 molecules, do they mean bound to TFIID or bound to DNA, or both or a mixture? Same for TAF4. If they refer to DNA bound, do they know if they are bound to DNA as part of TFIID or by themselves? With the overexpression, this is a little more difficult to deconvolute and to interpret cleanly.

This comment is no longer applicable.

For the residency time on DNA, how can the authors distinguish if the complex is bound to core promoter DNA or tethered to the +1 nucleosome via reader domains? And, again, is this occurring as individual subunits or in the context of the complex or a mixture? While the two different values for the two subunits make it less likely that both of them are binding as part of a full TFIID, the different possibilities should be kept in mind when interpreting the results.On that note, TAF4 has activator binding domains, so part of its dynamics can be affected by that, and this has to also be seen in the context of the overexpression

This comment is no longer applicable.

That fewer TAF7 binds to chromatin when TBP is depleted, but those bound stay longer, again needs to be put in the context of the overexpression and of the partial depletion of TBPIf the authors are assuming that TAF7 and TAF4 somehow act differently because they are present in different TFIID complexes, should the immuno-purification via TAF4 show substochiometric amounts of some of the other components? Can they check that?

This comment is no longer applicable.

Can the authors add a statement about how the CUT&Tag experiments and the single particle experiments relate to each other quantitatively?

This comment is no longer applicable.

(316) "based on the crystal structure of the TFIID complex" – should be "cryo-EM structure". On that note, there are two copies of TAF4, and based on structural studies, each of them can be close or far from TBP depending on whether TBP has or not bound the core promoter. By the way, TAF1 is even closer (directly interacting with TBP through its N-terminus) in free TFIID, but further that at least one copy of TAF4, once TBP engages the core promoter.

We agree and we have revised the Discussion section regarding TFIID structure. This new paragraph is now in page 14.

(327-8) "In contrast to TAF4, the largest subunit maintains its binding topromoters in the absence of TBP depletion." This does not make sense. Do they mean "in the case of TBP depletion"?

We agree and have modified the text to reflect this change “Therefore, TAF1 binding is unaffected by TBP-depletion" (pg. 11)

Reviewer #3 (Recommendations for the authors):– The authors should show and quantify the depletion of TBP in all cellular compartments, cytoplasm, nuclear soluble and the chromatin associated fractions after 6 and 16 hours of auxin treatment. In the Result section and in the legend of Figure 1A the authors should indicate which type of extract they are analysing on the western blot.

We agree and have performed cell fractionation experiments. We assessed mAID-TBP levels +/- IAA through Western blot analysis for each fraction. We show that TBP depletion is evident in both the cytoplasmic and chromatin bound fractions, which is consistent with the CUT&Tag data that primarily detects chromatin bound proteins. We have now included this cell fractionation experiment as a new panel in Figure1—figure supplement 1, and have referenced it in the main Results section. We have also clarified throughout the manuscript when the Western blots are performed on whole cell lysates.

– As the authors state that after the used auxin treatment about 10-20% TBP is remaining, the numerous statements giving the impression that TBP has no role in Pol II transcription should be tuned down throughout the whole manuscript. What if the remaining 10000-20000 TBP molecules can still work in Pol II transcription because they are very dynamic?

Previous biochemical studies, especially in vitro transcription experiments, have shown numerous evidence that TBP is necessary and sufficient for Pol II transcription. We agree that our system can only test if TBP is required/necessary for Pol II transcription in mESCs. We have revised the manuscript to specifically adhere to the conclusion that acute TBP depletion does not affect Pol II transcription, which we believe is justified by several orthogonal lines of evidence in our study. We agree that we cannot say that TBP has no role/function in Pol II transcription. Towards clarifying this in our manuscript, we also add in our Discussion section the following: “However, these results do not provide information on the functional role of TBP when recruited to Pol II promoters. One possibility is that Pol II transcription has evolved to have multiple redundancies such that removal of one factor (TBP) may not fully disrupt the whole system. Another possibility is that the transient nature of the embryonic cell state, with its unique chromatin landscape, enables Pol II transcription without TBP. Regardless, at the level of TBP depletion that we observe, we conclude that in mESCs Pol II transcription can occur independently of TBP” (pg 14-15).

Although it is possible, we currently have no evidence to suggest that the decreased levels of TBP after depletion would alter TBP-DNA binding dynamics. SPT analysis for endogenously tagged Halo-TBP has been published (PMID: 29939130) shows that TBP is one of the more stable-binding TFs imaged thus far, with average residence times on the order of minutes (~60-90 seconds). In contrast, most other TFs have residence times of seconds or less. Indeed, as this reviewer noted below, TBP dynamics is regulated by NC2 and BTAF1 complexes (Mot1 in yeast). These enzymes have been shown to be required to remove TBP from DNA, which is consistent with the long residence times of TBP. It is unclear how these enzymes would act under severely decreased TBP molecules. However, the Martianov 2002 paper (PMID: 12411709) has shown that TBP knockout embryos were still capable of generating newly transcribed RNA. Our molecular/genomics analyses using TBP depletion are consistent with this previous study, which we cited in our manuscript, and significantly advance the field by considering three potential mechanisms for TBP-independent Pol II transcription.

– How many TBP molecules work in Pol I, Pol II and Pol III GTF complexes? If there are many more TFIIIB complexes (containing TBP) than TFIID or SL1 complexes, it is logical that the first effect one will see after TBP depletion is the decrease of Pol III transcription. Can the authors quantify the number of TBP-containing TFIIIB complexes versus TFIID and SL1 complexes?

Absolute quantification of the number of molecules for TBP (and CTCF in the cited research article) was performed using rigorous Fluorescence Correlation Spectroscopy of endogenously tagged factors. To our knowledge, no similar study has been performed for Pol I, Pol II, or Pol III GTFs. Furthermore, this FCS approach quantifies molecules and cannot distinguish between different complex-bound molecules. Unfortunately, with the current limitations of the technology, we are unable to address this comment.

– Along the same lines, the authors should also test the effect of TBP depletion on Pol I transcription.

Please see our response to consensus comment #4 above.

– How dynamic is TBP in comparison to Pol I, Pol II and Pol III TAFs and/or Brf1? What are the corresponding DNA residency times of these factors?

As mentioned above, SPT analysis for endogenously tagged Halo-TBP has been published (PMID: 29939130), revealing long residence times compared to most other TFs. TAF4 and TAF7 have much shorter residence times compared to TBP, as measured by over-expression constructs. For a more rigorous approach, we would need to endogenously tag all factors the reviewer requested and perform the SPT analysis. This is currently not feasible in the time frame, but we are excited to pursue this line of research in the future.

– From Figure S2A it is evident that the most highly depleted genes for TBP binding are the most highly transcribed (high Pol II and NET-seq signals). Could TBP be more dynamic there? The same correlation is visible with TAFs (Figure 7D-F). Could TAFs favor TBP dynamics at promoters (making it more "depletable", maybe through TAF1 TAND)? Authors should show, side-by-side, the CUT&Tag occupancy heatmaps for TBP, TAF1, TAF4, Pol II and NET-seq with genes ordered by TBP occupancy (+/- IAA).

Genomics-based analyses such as our CUT&Tag and NET-seq data do not provide information on TBP dynamics. The higher genomics signal is only indicative of binding at the population level, which is correlated to the level of transcription. This observation is generally believed to be one of two reasons: (1) genes that are highly expressed have these factors bound for more time instances compared to less transcribed genes (bursting mechanism); or (2) at a population level, more cells are transcribing these genes while other cells are not (cell-to-cell variability). Therefore, it’s expected that highly expressed genes would have high Pol II signal, and high GTFs (TBP and TAFs) signal because their occupancy are highly correlated to transcription level.

It is also not surprising that the largest TBP depletion is most evident on promoters with high TBP occupancy, but the relative level of depletion is the same for promoters with high vs low occupancy, about 80-90% depletion. If highly expressed genes are “more depletable”, then the scatter plot in Figure 1E (TBP Ctl vs IAA) would look more exponential than a linear correlation. Instead, the r^2^ value for the linear regression analysis is very high, which means that highly expressed genes are experiencing similar levels of TBP depletion as middle/lowly expressed genes.

Furthermore, as we discussed in previous comments, the previously published residence time of TBP is a description of the average molecule and is independent of locus-specificity. That is, this data lacks information on whether TBP binding dynamics would change depending on which promoter it binds to. It is possible that highly expressed genes would also have more binding of NC2/BTAF1 but this data would still be correlational and would not provide direct evidence of differential TBP binding dynamics.

In the revised Figure 6—figure supplement 3D, we now include the side-by-side heatmaps for TBP, TAF1, TAF4, TFIIA, Pol II, and NET-seq as ordered by TBP occupancy.

– Figure Supp. 5G (not cited in the text) shows that under TBP and TBPL1 (TRF2) co-depletion Pol II occupancy increases about 5 fold (in replicate 2). How do the authors explain this increase? Could this be a normalization artefact? How is it possible that authors conclude from these conflicting replicate experiments that Pol II occupancy is not affected globally (lines 242-243)?

We have performed two additional replicates of Pol II CUT&Tag in TBP-depletion + TRF2 KO cells (new Figure 4—figure supplement 1I-K). For three of the four replicates, we find that Pol II signal is very similar between the TRF2 knockout and the double TBP-depletion + TRF2 KO. Therefore, we believe that the increase seen in replicate 2 (Figure 4—figure supplement 1J) is likely an artifact of normalization. We have included a citation to (now new) Figure 4—figure supplement 1J in the manuscript.

– Experiments described in Figure 6 are lacking several key controls before they can be interpreted. First the authors would need to demonstrate that ALL the "overexpressed" Halo-tagged TAF4 or Halo-TAF7 incorporate into endogenous TFIID complexes. Otherwise the presented Spot-On measurements are meaningless, because the authors may simply measure the behavior of free TAFs. What is the sense of deriving/showing the % of free/complexed/bound states in an overexpressed system (Figure 6)? No comparison can be made among different subunits (TAF7 vs TAF4) as attempted by the authors in the discussion (line 403, "our SPT results indeed show that TAF7 is much less DNA-bound..")

As discussed above with reviewer 2’s comments, we agree that our over-expression system is limited in capacity, and that endogenous tagging is the more rigorous approach. Since our imaging experiments are somewhat preliminary, we have removed these experiments from the manuscript.

– To normalize their CUT&Tag data, the authors used a spike-free normalization method. This method has only been validated for very abundant histone mark ChIPs, not for Pol II. It is not clear why the authors think that spike-free normalization method is needed for Pol II CUT&Tag normalization. In Supp. Fig1F, the authors show two Pol II CUT&Tag replicates with *E. coli* spike-in and the replicates are quite different. Why do the authors think that the normalized data showing Pol II decrease in IAA condition is not correct?

We agree that the *E. coli* spike-in normalized replicates are showing increased variability. We had aimed to use the ChIPseqSpikeInFree normalization method as an orthogonal system for validation. However, to gain more confidence in the results, we have performed two additional replicates of the Pol II CUT&Tag +/- IAA using S2 (*Drosophila*) cells as spike in control. We now include the heatmaps for these replicates normalized to S2 reads in the new Figure 1—figure supplement 2A and are referenced in the main text. Importantly, the S2-normalized data is consistent with the ChIPseqSpikeInFree normalization, and with our conclusions that Pol II transcription is largely unaffected by acute TBP depletion.

– How do authors interpret the widespread CUT&Tag signals they obtain on gene bodies for TAFs (especially TAF1) and partially also for TBP. Do these correspond to non-specific, transient binding events? If yes one would expect that a portion of the signal localized around promoter regions would also be of the same nature.

CUT&Tag uses pA-Tn5, which has an intrinsic preference for accessible DNA when integrating the adapters. Although the CUT&Tag protocol is designed to minimize this intrinsic preference, some of the background signals on our IgG CUT&Tag controls are located on accessible DNA (See new Figure 1—figure supplement 1F). The highest expressed genes have been shown to have lower nucleosome occupancy than mid-level expressed genes (PMID: 22085965 and others), and heat shock genes have been shown to be massively depleted of nucleosomes upon activation (PMID: 18614012). We speculate that the CUT&Tag signal that we observe within gene bodies could be a product of the pA-Tn5 integration spreading away from the TSS when the surrounding nucleosomes are disrupted.

Non-specific transient binding events could contribute to the CUT&Tag signal, but presumably, other genomics-based methods like ChIP-seq would also have a percentage of the signal corresponding to non-specific binding. Our IgG CUT&Tag controls would represent a measure of such non-specific/background binding levels. We have now included the heatmaps for the IgG controls in the new Figure 1—figure supplement 1F. With this control, it is evident how strong the specific binding is for our GTFs to the TSS.

– Heat-shock & RA induction might not be the best systems to test for the "reinitiation" hypothesis since TBP might already be promoter bound prior to activation. Indeed, there is a TBP peak even in non-treated condition in the tracks shown in Figure 3 and S3.

We agree that the HS response may not be ideal to test for reinitiation because it is well known to be regulated at the paused Pol II stage. We have added this statement in the text to reflect this change:

“The heat shock response is a classical gene induction system regulated at the level of paused Pol II elongation (ref), and HS genes may retain a scaffold of PIC components following promoter pausing release. Therefore, we also examined the role of TBP in an orthogonal gene activation system through retinoic acid (RA)-mediated differentiation of C64 mESCs, which leads to the silencing of pluripotency markers and activation of ectoderm-specific genes” (pg 8).

It is unclear in the literature whether the RA-induced genes are regulated through Pol II pausing, but at least some of the genes only have background level TBP CUT&Tag signal at the promoter region, including Cdx1 (Figure 3—figure supplement 1D). We have also examined the change in Pol II CUT&Tag levels relative to TBP CUT&Tag levels in control conditions (see scatter plots in Author response image 2).

For each gene, we calculated the total number of normalized Pol II or TBP reads +/- 1 Kb of the TSS. The left scatter plot shows all genes, and the right one shows all genes with increased Pol II CUT&Tag during RA and with TBP levels under 200 read counts. We highlighted several genes in red whose expression increased upon RA treatment but had very low TBP levels under control conditions.

– The RA treatment experiment has to be repeated similarly to the HS experiment, first 6h auxin treatment to degrade TBP and then RA treatment. At the beginning of the RA treatment TBP is still present in the cells making the interpretation of these experiments impossible.

We had originally planned to do this treatment (6 hours of IAA, then overnight IAA+RA). However, we found that 16-18 hours of IAA treatment is as much as we can do without significant cell death. At 24 hours of TBP depletion, we observe over 50% cell death. So we are limited with our IAA treatment timing to avoid secondary effects due to cell death. Unfortunately, RA treatment does not result in gene expression changes prior to overnight (16-18 hours) of treatment. Most published studies perform 24-48 hours before assessing changes in gene expression. We had done a preliminary test by performing RT-qPCR on two early RA-induced genes and observed no changes after 6 hours of RA treatment, but induction occurred after 16 hours. We believe we are at the cusp of the initial RA effects at 16 hours, and likely this is why we observe only about ~100 differentially expressed genes. Although IAA treatment started concurrently with RA treatment, we believe that the TBP degradation is near complete before the early RA genes are induced. We acknowledge that our compromise of simultaneous IAA+RA 16 hr treatment is not ideal, but we are also limited by the restrictions of the system.

– In several endogenous TAF IPs TBP has been found to be sub-stoichiometric, could this mean that TBP is not a stable component of mammalian TFIID complexes? TBP dynamics are also regulated by NC2 and BTAF1 complexes (PMID: 20627952), that could influence the interpretation of the obtained result.

In all previously published biochemical purification of TFIID that we are aware of, TBP is a stable component of the complex in human cells (for example PMID: 9882510) and in yeast cells (for example PMID: 2662184). In the study that the reviewer cites (“Chromatin interaction of TATA-binding protein is dynamically regulated in human cells” PMID: 20627952), the authors show that NC2 and BTAF1 regulate the DNA/chromatin binding dynamics of TBP, but not its complex formation with TFIID.

Regarding the reviewer’s comment “… that could influence the interpretation of the obtained result”, unfortunately we are unclear which interpretation of which result the reviewer is referring to. Is the reviewer referring to the IP-MS data (Figure 5), and our interpretation is that TFIID forms even when TBP is depleted? If so, we do not believe that our interpretation would be affected by the DNA/chromatin binding dynamics of TBP as regulated by NC2/BTAF1. Given that previous literature showing TBP is a stable component of TFIID through biochemical purification, and more recently that TBP forms stable ternary complex with TAF11 and TAF13 (PMID: 29111974), our initial prediction was that TFIID would not form when TBP is depleted, but the IP-MS data shows otherwise.

– The discussion sentence "The IP-MS data showing that the TFIID complex forms upon TBP depletion is surprising" is misleading, since at structural level TBP depletion is not expected to deeply affect any TFIID subunit or complex integrity.

Based on the reanalyses and new experiments recommended by all three reviewers, we have revised our Discussion section regarding TFIID. We believe that the discussion is now stronger as a result of the revisions.

– "The altered dynamics of TAF4 upon TBP depletion may be due to its close proximity to TBP, based on the crystal structure of the TFIID complex". There is no crystal structure of TFIID. TAF4 is instead involved in the interaction with TFIIA and, quite indirectly, with DNA-bound TBP.

We agree and have corrected this oversight in the revised manuscript.